# Genetic mapping reveals *Nfkbid* as a central regulator of humoral immunity to *Toxoplasma gondii*

Scott P. Souza[1,2], Samantha D. Splitt[1,2], Juan C. Sànchez-Arcila[1], Julia A. Alvarez[1,2], Jessica N. Wilson[1,2], Safuwra Wizzard[1], Zheng Luo[3], Nicole Baumgarth[3], Kirk D. C. Jensen[1,4] *

1 School of Natural Sciences, Department of Molecular and Cell Biology, University of California, Merced, Merced, California, United States of America, 2 Graduate Program in Quantitative and Systems Biology, University of California, Merced, Merced, California, United States of America, 3 Center for Immunology & Infectious Diseases, and Department of Pathology, Microbiology and Immunology, University of California, Davis, Davis, California, United States of America, 4 Health Science Research Institute, University of California, Merced, Merced, California, United States of America

* kjensen5@ucmerced.edu

**Data Availability Statement:** RNA-sequencing data generated in this study has been deposited in the NCBI Sequence Read Archive (Bioproject accession number PRJNA637442).

## Abstract

Protective immunity to parasitic infections has been difficult to elicit by vaccines. Among parasites that evade vaccine-induced immunity is *Toxoplasma gondii*, which causes lethal secondary infections in chronically infected mice. Here we report that unlike susceptible C57BL/6J mice, A/J mice were highly resistant to secondary infection. To identify correlates of immunity, we utilized forward genetics to identify *Nfkbid*, a nuclear regulator of NF-κB that is required for B cell activation and B-1 cell development. *Nfkbid*-null mice ("*bumble*") did not generate parasite-specific IgM and lacked robust parasite-specific IgG, which correlated with defects in B-2 cell maturation and class-switch recombination. Though high-affinity antibodies were B-2 derived, transfer of B-1 cells partially rescued the immunity defects observed in *bumble* mice and were required for 100% vaccine efficacy in bone marrow chimeric mice. Immunity in resistant mice correlated with robust isotype class-switching in both B cell lineages, which can be fine-tuned by *Nfkbid* gene expression. We propose a model whereby humoral immunity to *T. gondii* is regulated by *Nfkbid* and requires B-1 and B-2 cells for full protection.

## Author summary

Eukaryotic parasitic diseases account for approximately one fifth of all childhood deaths, yet no highly protective vaccine exists for any human parasite. More research must be done to discover how to elicit protective vaccine-induced immunity to parasitic pathogens. We used an unbiased genetic screen to find key genes responsible for immunity to the eukaryotic parasite *Toxoplasma gondii*. Our screen found *Nfkbid*, a transcription factor regulator, which controls B cell activation and innate-like B-1 cell development. Mice without *Nfkbid* were not protected against *T. gondii* and were deficient at making

**Funding:** This work was supported by NIH grants R01 AI137126, R15 AI131027, a Hellmann Fellowship and an MCB departmental award to KDCJ; and by NIH grants U19-AI109962 and R01 AI117890 to NB. The sequencing was carried out at the DNA Technologies and Expression Analysis Cores at the UC Davis Genome Center, supported by NIH Shared Instrumentation Grant 1S10OD010786-01. We would like to thank David Gravano and the UC Merced Stem Cell Instrumentation Foundry for their assistance designing panels and utilizing instruments, supported by the DoD Research and Education Program for HBCU/MSI Instrumentation Grant W911NF1910529. The funders had no role in study design, data collection and analysis, decision to publish, or preparation of the manuscript.

**Competing interests:** The authors have declared that no competing interests exist.

antibodies against the parasite. Our survival studies of vaccinated mice with and without B-1 compartments found that B-1 cells improved survival, suggesting that B-1 cells act in conjunction with B-2 cells to provide vaccine-induced immunity. *Nfkbid* and other loci identified in our unbiased screen represent potential targets for vaccines to elicit protective immune responses against parasitic pathogens.

## Introduction

The goal of vaccination is to induce immunological memory that can protect from natural infection challenge. Depending on the pathogen, effective memory would need to protect also against a wide variety of pathogen-specific strains encountered in nature. Such protection is termed heterologous immunity and is effective against pathogen strains that differ in virulence, immune evasion, or polymorphic antigens. Parasites represent a special challenge to vaccine development. Indeed, an entirely protective vaccine has yet to be achieved for any human parasite [1]. The apicomplexan parasite *Toxoplasma gondii* provides an excellent system to explore requirements for heterologous immunity to a parasitic pathogen. *T. gondii* is a globally spread intracellular protozoan parasite of warm-blooded animals that exhibits great genetic diversity [2]. *T. gondii* strains differ dramatically in primary infection virulence in laboratory mice [3] and in severity of human toxoplasmosis [4–6]. Such infections can be overcome by immunological memory responses elicited by vaccination or natural infection. In particular, memory CD8 T cells and induction of IFNγ are primarily responsible for protection against lethal secondary infections with the widely studied type I RH strain, which has a lethal dose of one parasite in naïve mice [7–9]. CD4 T cells are required to help the formation of effector CD8 T cell [10] and B cell responses [11], but the ability to adoptively transfer vaccine-elicited cellular immunity to naïve recipients against type I RH challenge is unique to memory CD8 T cells [8,9].

The role of B cells in *T. gondii* infections is less understood. Previous studies showed that B cell deficient mice (muMT) are extremely susceptible to primary [12], chronic [13] and secondary infections [14], despite unimpaired levels of IFNγ. Passive transfer of antibodies from immunized animals into vaccinated muMT mice significantly prolongs their survival after challenge [11,14]. IgM seems particularly suited for blocking cellular invasion by *T. gondii* [15], while IgG can perform both neutralization [16] and opsonization functions [17]. Antibody responses against *T. gondii* are dependent on CD4 T cells [11,18], and are regulated by cytokines that modulate T follicular helper cell and germinal center B cell formation in secondary lymphoid organs [19], suggesting conventional "B-2" B cell responses provide antibody-mediated immunity to *T. gondii*.

In addition, "B-1" cells are innate-like lymphocytes that are known for producing self- and pathogen-reactive "natural" IgM. B-1 cells are the predominant B cell compartment within the body cavities, including the peritoneal and pleural spaces and contribute to antigen-specific responses to many pathogens. In mouse models of secondary bacterial infections, including *Borrelia hermsii*, *Streptococcus pneumoniae* and non-typhoid *Salmonella*, vaccination induces protective memory B-1 cells to T cell-independent bacterial antigens (reviewed in [20]). This memory is often restricted to the B-1(b), or CD5- subset of B-1 cells [21], but not in all models [22]. In the *T. gondii* model, one study suggested that primed CD5+ B-1(a) cells can rescue B-cell-deficient mice during primary infection with a low virulence strain [12]. Memory B cells are also appreciated to secrete pathogen-specific IgM [23], and generate somatically mutated IgM to combat blood stage secondary infection with *Plasmodium* [24]. Whether IgM

responses to *T. gondii* are B-2 or B-1 derived is unknown. Moreover, the role of B-1 cells in promoting immunity to *T. gondii* during a secondary infection has yet to be determined.

Particularly troubling for vaccine development is the lack of sterilizing immunity achieved following *T. gondii* infection. Unlike the highly passaged lab type I RH strain, the less passaged type I GT1 strain and atypical strains, many of which are endemic to South America, cause lethal secondary infections in C57BL/6J mice and co-infect (i.e. "superinfect") the brains of challenged survivors [25]. Vaccinated C57BL/6J mice are also susceptible to challenge with homologous strains, suggesting that defects in antigen recognition cannot fully explain immune failure to clear virulent strains [25,26]. During secondary infection memory CD8 T cells become exhausted, but checkpoint blockade fails to reverse disease outcome [27]. The data suggest yet unknown mechanisms are needed to provide heterologous immunity to highly virulent strains of *T. gondii*. Therefore, we set out to address whether additional requirements are necessary for immunity to *T. gondii*. Through use of forward and reverse genetics, we discovered a previously unidentified essential role for *Nfkbid* in immunity and antibody responses to *T. gondii*, and present evidence that both B-1 and B-2 cells assist resistance to secondary infection with highly virulent parasite strains.

## Results

### Non-MHC loci control resistance to secondary infection with *Toxoplasma gondii*

When mice are given a natural infection with a low virulent type III CEP strain and allowed to progress to chronic infection for 35–42 days, the immunological memory that develops is known to protect against secondary infections with the commonly studied lab strain, type I RH. In contrast, highly virulent "atypical" strains such as those isolated from South America (VAND, GUY-DOS, GUY-MAT, TgCATBr5) or France (MAS, GPHT, FOU) and the clonal type I GT1 strain led to morbidity and mortality in C57BL/6J mice at varying frequencies, depending on the strain type and their virulence factors [25]. To explore whether host genetics influenced the ability to survive secondary infection, a similar experiment was performed with A/J mice. In contrast to C57BL/6J mice, A/J mice were resistant to secondary infection with all virulent atypical *T. gondii* strains analyzed (Fig 1A) and cleared parasite burden as early as day 4 post challenge (Fig 1B). As is the case for laboratory mice of the *Mus musculus domesticus* sub species [25,28,29], in the naïve state A/J mice succumb to primary infections with atypical strains, such as VAND, GUY-DOS, and GUY-MAT (S1 Fig). These results suggest that at least one or more genetic loci control secondary infection immunity to virulent challenge.

A/J mice are known to be resistant to primary infections with the intermediate virulent type II strain [30], and prevent cyst formation due to polymorphisms in their MHC class I H-2 molecule, L$^d$ [31]. To test whether the H-2 locus contributes to immunity we compared secondary infections in C57BL/10 (B10) and C57BL/10.A (B10.A) mouse strains, the latter carrying the MHC H-2 locus of A/J (H-2a) in place of C57BL/6's MHC H-2 (H-2b). Compared to mice with the H-2b haplotype, mice expressing H-2a had less cysts and weighed more during chronic infection with type III strains (S2 Fig). Despite their relative health at the time of challenge, B10.A mice were highly susceptible (0%-30% survival) to secondary infection with certain atypical strains (VAND, GUY-DOS, GPHT, FOU), but displayed varying degrees of resistance to others (TgCATBr5, MAS, GUY-MAT; 60–100% survival) (Fig 1A). In the case of MAS secondary infection, B10.A but not B10 mice were highly resistant and exhibited reduced parasite burden by day 8 post challenge (Fig 1B). Together, the data suggest that while the MHC H-2a locus is an important modifier of resistance to certain *T. gondii* strains, this is not

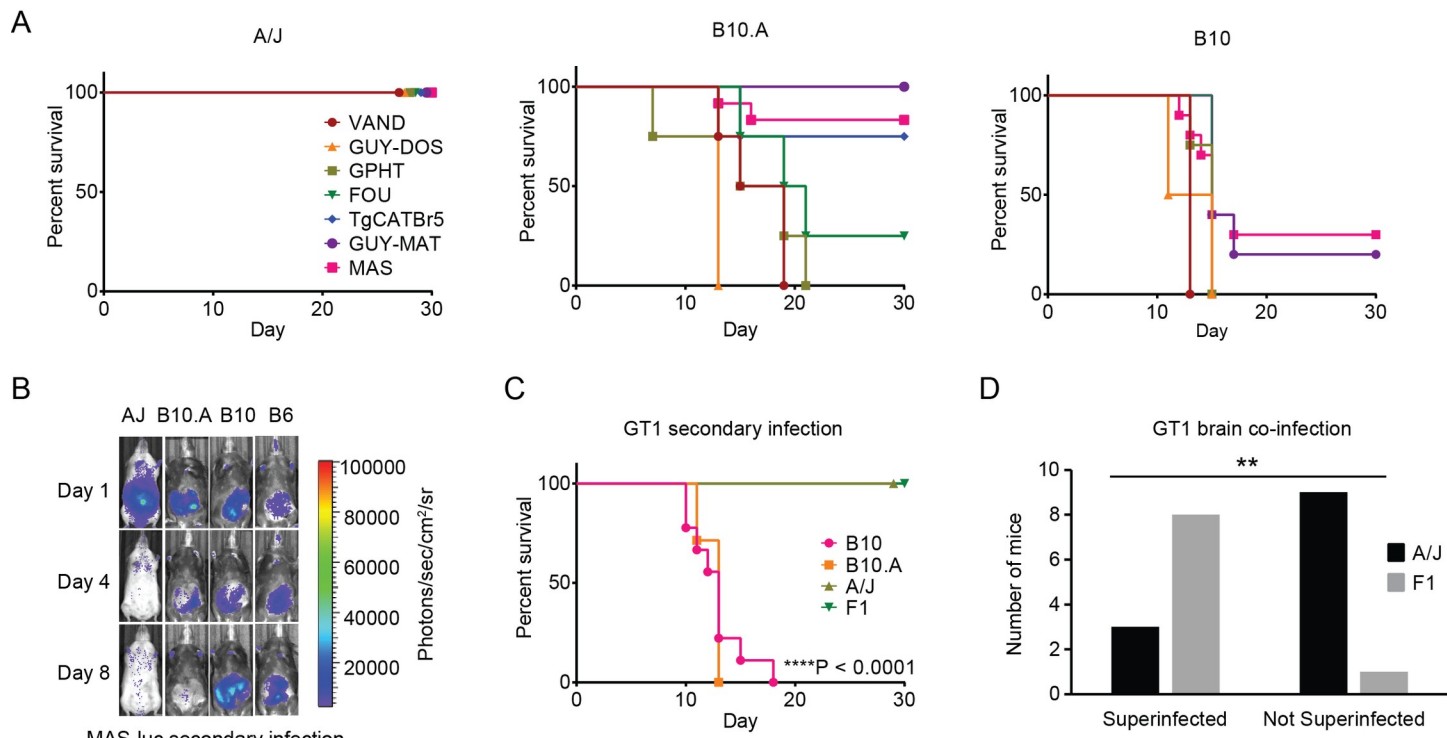

**Fig 1. MHC and non-MHC alleles promote immunity to virulent *Toxoplasma gondii* strains in A/J mice.** All mice were infected with $10^4$ type III CEP *hxgprt*-avirulent *T. gondii* parasites and allowed to progress to chronic infection; then, 35 days later, mice were challenged with $5x10^4$ of the indicated strains of *T. gondii*. A) Survival of A/J, C57BL/10J (B10), and C57BL/10J.A (B10.A) mice following secondary infection with atypical strains of *T. gondii*. Cumulative results are plotted from 1–2 experiments; n = 4 to 12 mice per parasite strain and mouse genetic background. B) Bioluminescence imaging of individual mice challenged with luciferase expressing MAS strain on days 1, 4 and 8 of secondary infection; parasite burden is shown as a heat map depicting the relative number of photons (photons/sec/cm2/sr) detected over a 5 minute exposure. C) Survival of B10 (n = 5), B10.A (n = 5), A/J (n = 12), and F1 (A/J x C57BL/6, n = 9) mice following secondary infection with the type I GT1 strain; ****P<0.0001, log-rank (Mantel-Cox) compared to A/J mice. Cumulative data from 1 to 2 experiments are plotted. D) Superinfection in surviving A/J (n = 12) and F1 (n = 9) mice following 35 days of secondary infection with the type I GT1 strain. To evaluate superinfection, brain homogenate was grown in MPA-xanthine selection medium, which selects for GT1 parasites expressing the endogenous *HXGPRT* locus and against the *hxgprt*- type III CEP strain used to induce chronic infection. Plotted is the number of mice for which the presence or absence of the GT1 strain was detected; ** P< 0.0075, Fisher's exact test.

true for every challenge. Importantly, the A/J genetic background encodes additional non-MHC-linked genes that control immunity to *T. gondii*.

## Genetic mapping reveals four loci that correlate with immunity to *Toxoplasma gondii*

To identify non-MHC loci that promote resistance to secondary infection, we first analyzed the outcome of infection with the type I GT1 *T. gondii* strain, as this strain caused lethal secondary infections in B10 and B10.A, but not in A/J or first filial generation A/J x C57BL/6J mice ('F1') (Fig 1C). Following secondary infections, A/J and F1 mice showed no overt symptoms of weight loss, dehydration, or lethargy. However, sterile immunity was not achieved. For example, the GT1 strain was present in the brains of these survivors (i.e. "superinfection"), and at greater frequencies in F1 compared to A/J mice (Fig 1D). Superinfections were also detected at high frequencies in B10 and B10.A surviving mice challenged with atypical strains. Overall, mice of the C57BL genetic background were more prone to superinfection compared to A/J mice (S1 Table). It is unknown whether virulent strains of *T. gondii* have evolved to superinfect hosts with immunological memory, as previously hypothesized [25]. Nonetheless, our results underscore the difficulty in achieving sterile immunity to parasites in otherwise genetically resistant hosts.

We then performed secondary infection experiments with the type I GT1 strain using 26 recombinant inbred (RI) mice (S2 Table). The AxB:BxA RI mouse panel contains an assortment of homozygous A/J and C57BL/6 alleles, which assist genetic mapping of loci that contribute to various phenotypes, including those related to *T. gondii* infection [30,32]. Genetic mapping revealed four distinct Quantitative Trait Loci (QTL) peaks with logarithm of the odds (LOD) scores greater than 3 on chromosomes 7, 10, 11 and 17 (Fig 2A). None of the QTLs bore evidence for epistatic interactions, and only the chromosome 10 QTL surpassed genome-wide permutation testing (n = 1000, P<0.05). Nevertheless, an additive-QTL model including all four QTLs best fits the data compared to any lesser combination of them (P<0.02, ANOVA). The estimated effect on the phenotypic variance observed in the RI panel is 24%, 41%, 21% and 27% for the chromosome (chr) 7, 10, 11 and 17 QTLs, respectively. Consistent with these estimates, complete phenotypic penetrance (i.e. allelic correlation at 100%) was not observed for any locus (Fig 2B). Moreover, replacing chromosomes 7 or 10 of C57BL/6J with those of A/J conferred no survival advantage to secondary infection in consomic mice (S3A Fig). Regardless, small effect QTLs controlling complex traits can still lead to the identification of causal genes within a QTL region, as occurred for the successful identification of MHC 1 L$^d$ as the host resistance factor to chronic *T. gondii* infection [31].

*Nfkbid* is one of the most polymorphic genes within the chr7 QTL region (Mb 30.3–33.0) and sits between the genetic markers that flank the highest imputed LOD score (S1 Dataset). This gene encodes IκBNS, which is a member of atypical NF-κB inhibitors, called nuclear IκBs, reflecting their restricted cellular localization. Unlike classical inhibitors of NF-κB, atypical NF-κB inhibitors can modulate NF-κB to induce or repress transcription [33]. Previous work has shown that *Nfkbid* null mice completely fail to develop B-1 cells, lack circulating IgM and IgG3 antibodies, and cannot respond to T-independent (T-I) type II antigens such as NP-ficoll [34–36]. IκBNS also promotes early plasma blast differentiation [37] and IgG1 responses to model T-dependent antigens [38], enhances T cell production of IL-2 and IFNγ [39], supports development of T regulatory cells [40] and suppresses TLR-induced cytokine expression in macrophages [41]. The tetraspanin, *Tspan8*, is within the chr10 QTL (Mb 115.8–116.2) and is the most polymorphic gene between A/J and C57BL/6J mice in this region (S1 Dataset). *Tspan8* is 6-fold more highly expressed in spleens from A/J compared to C57BL/6 mice (immgen.org). *Tspan8* promotes cancer metastasis [42] and can impact leukocyte migration [43], but its role in immunity is largely unknown. Other polymorphic gene candidates within the four QTLs are listed in S1 Dataset.

## *Nfkbid* on chromosome 7 is required for immunity and the generation of *Toxoplasma gondii*-specific antibodies

Given the role of *Nfkbid* in several immune functions, degree of polymorphism and central location within the chr7 QTL, the requirement for *Nfkbid* in immunity to *T. gondii* was further explored. *Nfkbid* null *bumble* mice (C57BL/6) have previously been described, which were derived from an ENU mutagenesis screen. These mice possess a premature stop codon in *Nfkbid*, rendering them unable to support T-I antibody responses and B-1 development [35]. *Bumble* mice survived primary infection with the low virulent CEP strain at frequencies similar to wildtype C57BL/6J mice (Fig 2C) but succumbed to secondary infection three days earlier when challenged with the GT1 strain (Fig 2D). Since the C57BL genetic background is uniformly susceptible to GT1 secondary infections used in our genetic screen, susceptibility to challenge with the commonly studied type I RH strain was explored, which is normally controlled in vaccinated or chronically infected mice [7–9]. Importantly, *bumble* mice were entirely susceptible to secondary infection with the type I RH strain (Fig 2E), which exhibited

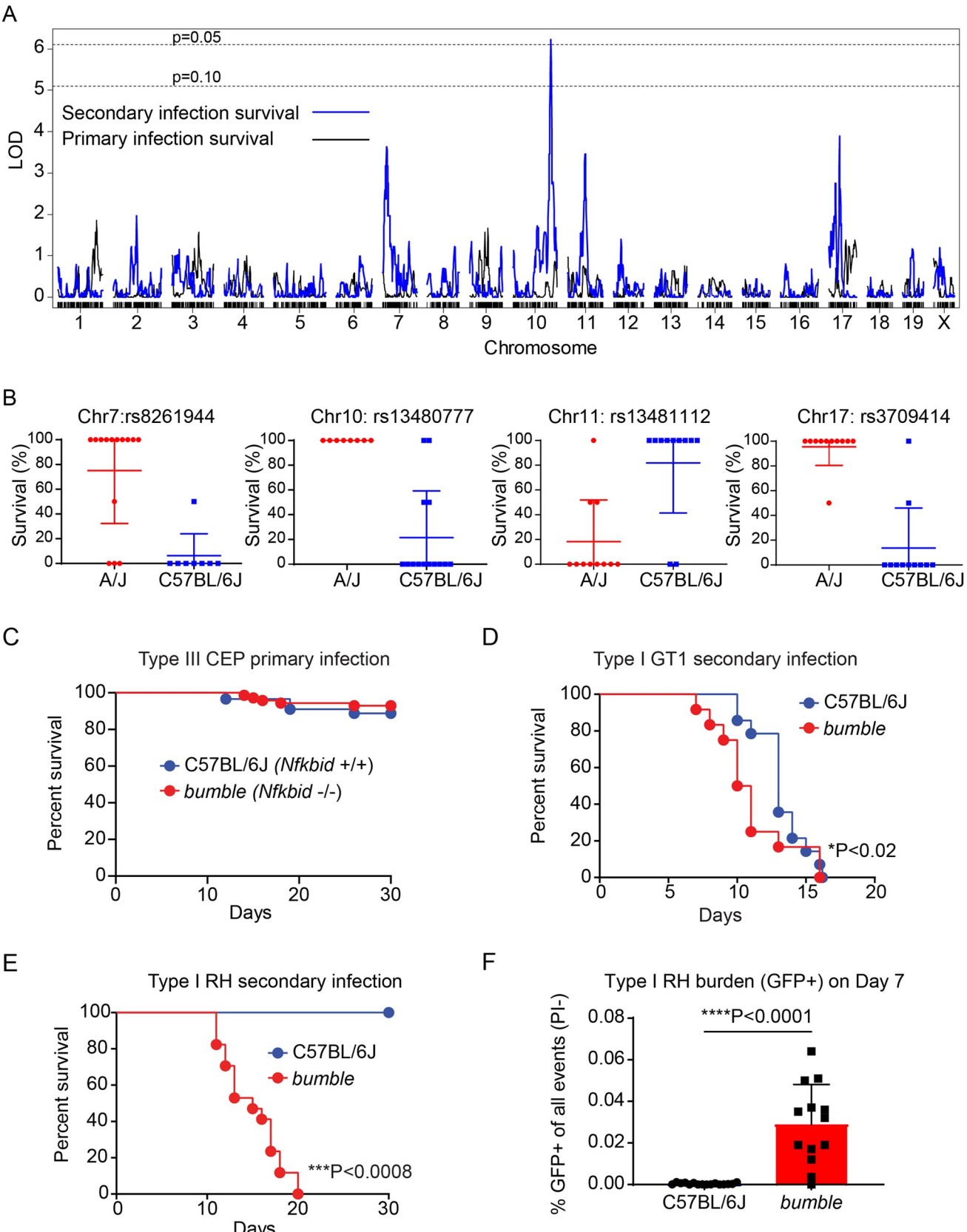

**Fig 2. Genetic mapping reveals *Nfkbid* is required for immunity to *Toxoplasma gondii* secondary infections.** 26 recombinant inbred (RI) mouse strains from the AxB;BxA panel were primed with $10^4$ type III *T. gondii* CEP *hxgprt-* parasites; then, 35 days later, mice were challenged with $5x10^4$ virulent type I GT1 *T. gondii* parasites (n = 2 per RI line). A) LOD scores for each marker were calculated using Haley-Knott regression and the running LOD scores of primary (black) and secondary infection survival (blue) for each genetic marker are plotted. 1000 permutations were performed to obtain the genome wide threshold LOD values; P = 0.10 and 0.05 thresholds are shown. B) Effect plots for the genetic markers closest to the maximal LOD scores calculated for the chromosome 7, 10, 11 and 17 QTLs are shown. Each dot indicates the percent survival of a unique RI line and whether it encodes an A/J (red) or C57BL/6 (blue) allele at the specified genetic marker. C) Cumulative survival of *bumble* (*Nfkbid-/-* C57BL/6) (n = 71) and wildtype (C57BL/6J) (n = 89) naïve mice infected with the avirulent type III CEP strain. D) Survival of chronically infected *bumble* (n = 12) and wildtype (n = 8) mice given a secondary infection with the type I GT1 strain. Cumulative survival from 3 separate experiments are shown; * P<0.02, Gehan-Breslow-Wilcoxon test. E) As in B, but survival to secondary infection with the type I RH strain is shown. Cumulative survival from 3 separate experiments is plotted (*bumble* n = 17, C57BL/6J n = 4); ***P<0.008, Mantel-Cox test. F) Frequency of GFP+ events in the peritoneal lavage 7 days post-secondary infection with GFP-expressing type 1 RH strain (RH 1–1). Each dot represents the result of one mouse, and cumulative results are shown from 3 separate experiments (*bumble* n = 13, C57BL/6J n = 16); ****P<0.0001, unpaired two-tailed t-test.

greater parasite loads compared to wildtype mice (Fig 2F). Moreover, *bumble* mice failed to generate parasite-specific IgM, and were poor producers of parasite-specific IgG3, IgG2b and IgG2a antibody responses after chronic infection (Fig 3A). The remaining antibodies that were secreted exhibited defects in their ability to block parasite invasion of host cells (Fig 3B). Antibodies from naïve mice fail to bind *T. gondii*, thus natural antibodies do not recognize *T. gondii*, consistent with previous reports [15]. Although *Nfkbid* promotes T cell production of IFNγ and IL-2 in in vitro stimulation assays [39] and promotes thymic development of FOXP3 + T regulatory cells [40], no impairment of T cell cytokine production was observed, nor were frequencies of FOXP3+ CD25$^{hi}$ CD4+ T regulatory cells altered in *bumble* compared to wildtype mice during chronic and secondary infection (S4 Fig). Moreover, cyst burden in the brain was the same between *bumble* and wildtype mice (S4 Fig), results which are consistent with unimpaired T cell responses required for cyst removal and preventing reactivation [44–48].

To determine where the breakdown in the B cell response occurred in *bumble* mice, immunophenotyping was employed. Consistent with previous reports, *bumble* mice have greatly reduced marginal zone B cells in naïve mice and did not increase in frequency during *T. gondii* infection (S5 Fig). Atypical B cells (FCLR5+ CD80+ CD73+), which respond to *Plasmodium* infections in mice [49], were also reduced in *bumble* compared to B6 mice following *T. gondii* infection (S5 Fig). The memory CD73+ B cell compartment in *bumble* mice bore evidence for reduced class switching during *T. gondii* infection, as they remain mainly IgM+IgD+ while C57BL/6J mice have higher frequencies of IgM-IgD- cells (Fig 3C). Most pronounced, however, is a large accumulation of transitional stage immature B-2 cells in *bumble* mice that occurs during chronic infection, implicating that B cell responses to *T. gondii* may require reinforcement from recent bone marrow derived emigrants that is blocked in the absence of *Nfkbid*. Hence, *Nfkbid* is not only an important regulator of B-1 cell development through the transitional stage of immature B cell development [50], but also for B-2 cell maturation, differentiation and activation during *T. gondii* infection.

## Defective B-1 and B-2 responses underlie *bumble's* defect in immunity

The impaired ability of *bumble* mice to generate parasite-specific antibodies, combined with the documented collapse of the B-1 cell compartment in *Nfkbid*-deficient mice [34,35], prompted us to directly assess the role of B-1 mediated immunity and humoral responses to *T. gondii*. First, total peritoneal exudate cells (PerC) from wild type mice were adoptively transferred into *bumble* mice at day 2 after birth, which allows optimal B-1 engraftment and self-renew for the life of the animal [51]. Then, *bumble* mice that received total PerC transfers were infected with the avirulent type III strain at 6–7 weeks of age and given a secondary infection 35 days later with the type I RH strain (Fig 4A). *Bumble* mice that received PerC partially reconstituted serum IgM to ~40% of wildtype levels (S6A Fig), consistent with previous studies

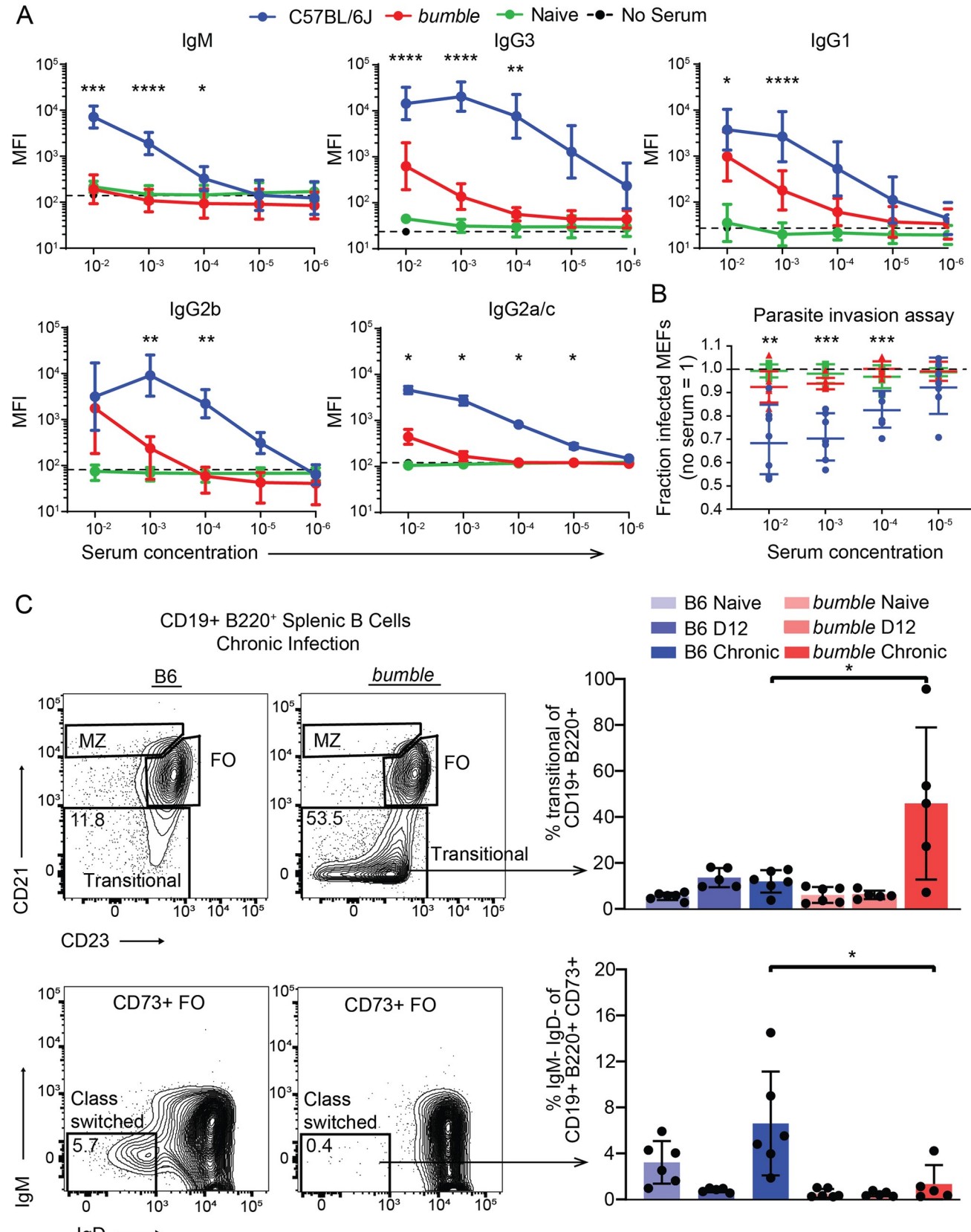

**Fig 3. *Nfkbid* is required for humoral responses to *Toxoplasma gondii*.** A) Whole fixed GFP+ parasites were incubated with serum from chronically infected mice, stained with fluorescently labeled anti-isotype antibodies and assessed by flow cytometry. Quantification of *T. gondii*-specific antibody isotype binding (IgM, IgG1, IgG2a/c, IgG2b, and IgG3) over a range of serum concentrations is shown. Background staining in the absence of serum is indicated by the dotted line for each isotype. Plotted is the cumulative average +/-SD of the geometric mean fluorescence straining intensity (MFI) from 3 separate experiments (*bumble* n = 8, C57BL/6J n = 11). B) Neutralization of GFP+ parasites coated with serum over a range of concentrations from the indicated chronically infected mice. Parasites were incubated in heat inactivated serum for 20 minutes before infection of mouse embryonic fibroblasts (MEFs) and assessed by FACS 2h later. The fraction of infected host cells (GFP+ cells) is normalized to that of parasite infections without serum. Each dot represents the serum from an individual mouse and cumulative results from 3 separate experiments are shown (*bumble* n = 8, C57BL/6J n = 7, naïve n = 6). For A and B, significance was assessed by unpaired t-tests with Holm-Sidak correction for multiple comparisons; **** P<0.0001, *** P< 0.001, ** P<0.01, * P<0.05. C) Representative FACS plots and the frequency of splenic transitional B-2 cells and IgM- IgD- CD73+ conventional memory B cells at naïve, d12 of primary infection, and chronic infection in *bumble* and C57BL/6J mice. Cumulative data from two experiments, n = 5–6 mice/condition is plotted; * P<0.05 by unpaired two-tailed t-test.

[51], and had a significantly delayed time to death relative to non-transferred littermates following type I RH challenge (Fig 4A). Previous studies have shown B-1 cells respond to infections and create both pathogen- [21,52] and microbiota-specific antibodies [53]. Antibody profiling of *bumble* mice that received PerC transfer showed a trend of increased anti-*T. gondii* antibody generation following chronic infection (Fig 4B), but antibody reactivity to the parasite did not reach levels observed in wildtype mice, suggesting the B-1 compartment has a limited role in generating high-affinity parasite-specific antibodies. The B cell compartment responsible for generating parasite-specific antibody was further confirmed with Igh-allotype chimeric mice which allow tracking of IgM responses of allotype-marked B-1 and B-2 cells [54]. In this experimental setup, endogenous B cells of the C57BL/6 background (IgH-b allotype) are depleted with allotype specific anti-IgM-b antibodies and replaced with transferred PerC B-1 of the IgH-a allotype which are refractory to the depletion antibodies and will engraft for the life of the animal. Following removal of the depleting antibodies, the endogenous B-2 cell population reemerge and are marked with anti-IgM-b antibodies, while the transferred B-1 cells are marked with anti-IgM-a antibodies (Fig 4C). Assessing antibody responses generated in IgH-allotype chimeric mice during a *T. gondii* infection revealed the presence of B-1 derived IgM-a antibodies that had low reactivity to *T. gondii* at days 14 and 30 post-infection, but the majority of highly reactive parasite-specific IgM-b was derived from the B-2 compartment (Fig 4D).

Attempts to explore the role of B-1 and B-2 cells utilizing B cell deficient muMT mice were complicated by their high susceptibility to primary infection with the CEP strain, irrespective of whether they received PerC as neonates or splenic B-2 cells prior to the primary infection (S6C–S6E Fig), thus underscoring the importance of B cells in resistance to *T. gondii* infection [13]. Instead, mixed bone marrow chimeras were generated in which irradiated *bumble* or wildtype recipients were transferred wildtype or *bumble* bone marrow to reconstitute B-2 cells and the rest of the hematopoietic compartment. Since B-1a cells do not efficiently reconstitute irradiated recipients from adult bone marrow [55], some recipients received wildtype PerC to restore the B-1 compartment in this setting [52]. These and other bone marrow chimeras all succumbed to primary CEP infections (S7 Fig). To bypass the susceptibility of irradiated bone marrow recipients to live *T. gondii* infections [56], bone marrow chimeras were vaccinated with a replication deficient uracil auxotroph strain (RH *Δompdc Δup*) improving overall survival (S7 Fig). Whereas vaccinated wildtype recipient mice that received *Nfkbid*-sufficient but not *bumble* bone marrow were able to survive type I RH challenge, complete immunity was conferred only when they were transferred PerC (Fig 4E). *Nfkbid* is also important in the non-hematopoietic lineage, as complete immunity was restored only in wildtype but not *bumble* recipients (Fig 4E). In summary, our reconstitution studies emphasize the importance of *Nfkbid* sufficiency in multiple compartments and highlight B-1 cells as an important contributor to *T. gondii* immunity.

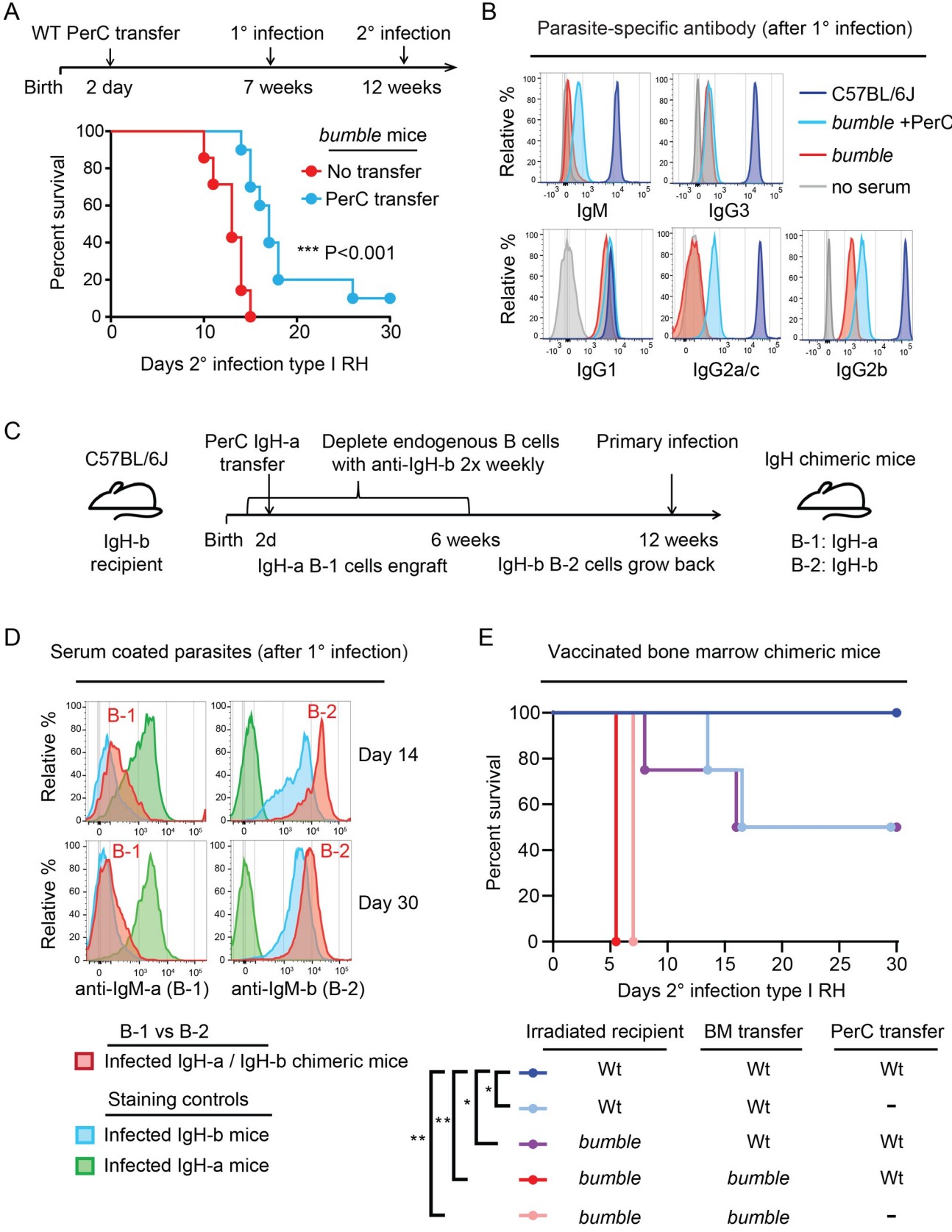

**Fig 4. The contribution of B-1 and B-2 cells to *Toxoplasma gondii* immunity in C57BL/6J mice.** A) Schematic of the secondary infection experiment using PerC reconstituted *bumble* mice. 2–4 day old *bumble* neonates were transferred $5x10^6$ total PerC and allowed to rest for 7 weeks before primary infection with the type III CEP strain. 5 weeks post-primary infection, mice were given a secondary infection with the type I strain RH. Survival of *bumble* mice given total PerC transfers (n = 10) relative to littermate controls (n = 12). Cumulative survival is shown from 3 separate experiments; *** P <0.001, Mantel-Cox test. B) Representative histograms of anti-isotype staining of parasites coated in serum ($10^3$ dilution for IgG, $10^2$ for IgM) from chronically infected *bumble*, *bumble* given PerC transfer, and WT mice. C) Schematic of neonatal allotype chimera generation. C57BL/6J neonates were given anti-IgHb to deplete endogenous B cells at day 1 post birth and twice weekly after for 6 weeks, thereby depleting the endogenous B-1 pool for the life of the animal due to their restricted fetal/neonatal window of development. Neonates were given $5x10^6$ total PerC from 6–8 week old IgH-a congenic C57BL/6 mice donors. These mice then rested for 6 weeks after the last depletion treatment to allow reemergence of the endogenous B-2 IgH-b cells. D) Representative histograms of serum derived anti-IgM-a (B-1 derived) or anti-IgM-b (B-2 derived) staining profiles of type I RH GFP+ parasites taken from the IgH allotype chimeras on day 14 and 30 following primary type III CEP infection. Staining controls with serum from chronically infected C57BL/6J IgH-b littermates, or IgH-a mice are shown. E) Irradiated *bumble* and WT recipients (45.1 or 45.2) were given WT or *bumble* bone marrow (BM) with or without total WT PerC (45.2). Mice were vaccinated with a replication deficient type I strain (RH *Δup Δompdc*) and 30d later challenged with type I RH. Cumulative survival is shown from 2–3 experiments (n = 4–9 per condition); * P<0.05, **P<0.01 by Mantel-Cox test.

## Evidence for enhanced B-1 and B-2 cell activation in resistant A/J mice

Since *Nfkbid* has a profound effect on the maturation and activation of multiple B cell populations in the C57BL/6J background, we extended our analysis of the humoral response to the resistant A/J background. A/J mice were found to have a superior IgG response to parasite lysate antigen during secondary infections (Fig 5A), suggesting enhanced humoral responses in this background. Indeed, both neutralization and opsonization potentials of immune sera from A/J mice were superior to those from C57BL/6J mice (Fig 5B and 5C). Within the spleen, both susceptible C57BL/6J and resistant A/J mice produced similar frequencies of memory B cell CD73+ FCRL5- and atypical memory CD73+ FCRL5+ CD80+ populations during infection, however in A/J mice these compartments were significantly increased in their class-switch recombination frequencies during secondary infection (Fig 5D). The enhanced class switch potential of resistant mice was also observed in B-1 cells. Within the peritoneal B-1 cell compartment (CD19+ B220^int CD11b+) increased percentages of CD5- B-1b cells were observed during secondary infection in A/J mice, which appear to have downregulated their BCR as evidenced by being IgM^lo (Fig 6A and 6B). To further validate these findings, frequencies of class switched IgM-IgD- CD5+ (B-1a) or CD5- (B-1b) cells within the CD43+ B-1 cell compartments of the peritoneum (S8 Fig) and spleen were determined, and a similar trend was observed (Fig 6C). Following infection, B-1 cells in the spleens of A/J mice maintained high levels of BAFFR and TACI expression and expressed higher levels of surface CD138 relative to C57BL/6J mice (Fig 6D and 6E), markers known to be induced by *Nfkbid* and important for B cell activation and differentiation into antibody secreting cells [37]. In addition to differences in humoral immunity, an increase in peritoneal CD8 T cell in vitro recall production of IFNγ was noted in A/J relative to C57BL/6J mice, however other parameters, including granzyme B and IL-2 expression and CD4 T cell responses were similar (S9 Fig). Collectively these data suggest that while CD8 T cell production of IFNγ is enhanced in resistant mice, and likely contributes to their resistance, B-2 and B-1 cells participate in a strikingly enhanced humoral response in A/J mice, offering insight into how immunity against parasitic infections may be achieved.

## B-1 cells in resistant A/J mice have enhanced germline transcription of *Ighg* constant regions, class switch recombination, and different activation profiles

The diminished antibody response in *bumble* mice as well as the B cell phenotypic differences noted in A/J mice prompted us to investigate further how *Nfkbid* may be modulating the B cell compartment in both genetic backgrounds. A transcriptomic approach was taken to define

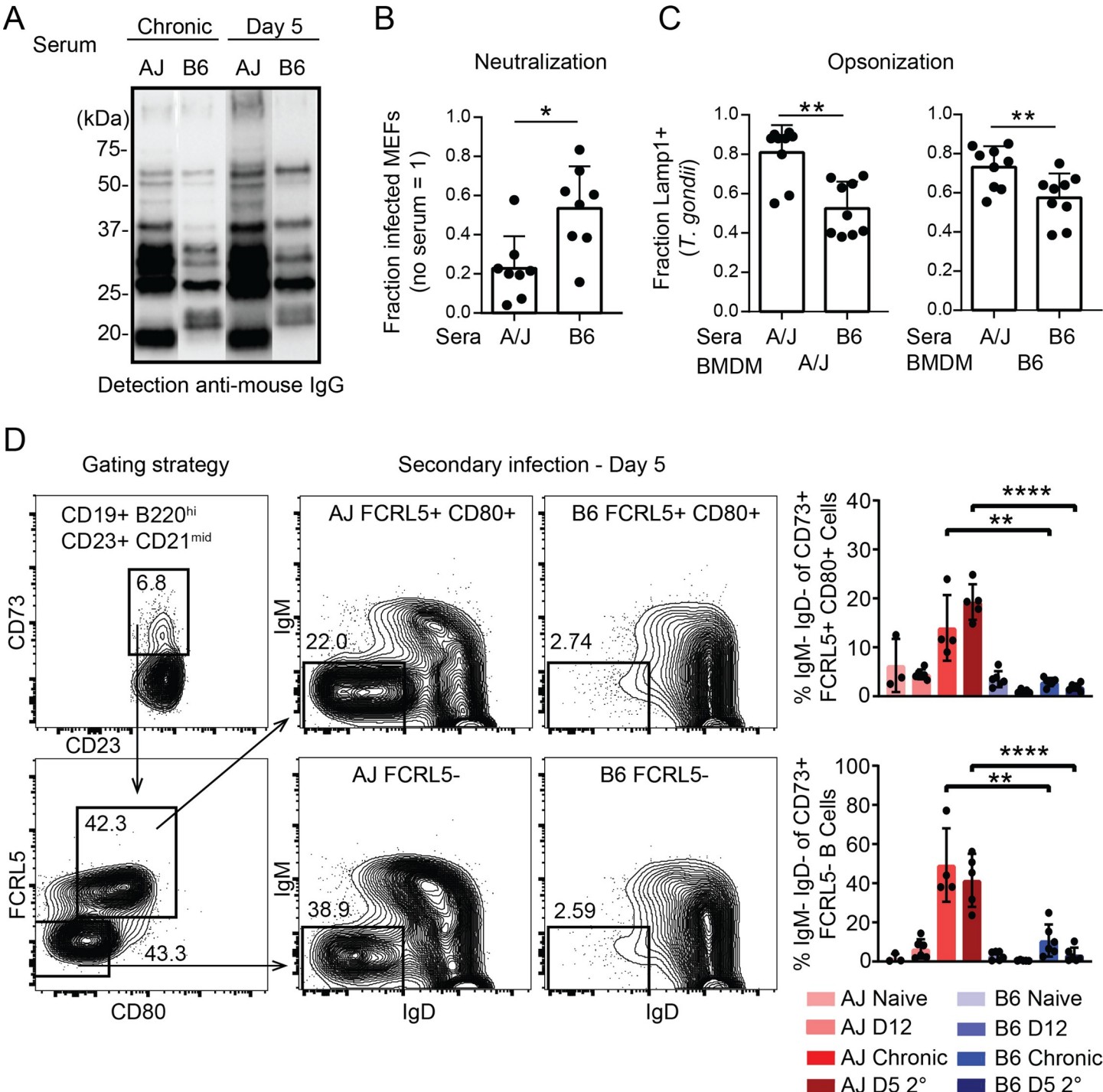

**Fig 5. Immunity in A/J mice correlates with enhanced class-switching in memory B-2 cells and increased serum reactivity to parasite lysate antigen.** A) Sera obtained from A/J and C57BL/6J (B6) mice chronically infected (CEP) or at D5 of secondary infection (GT1) were used to probe GT1 parasite lysate separated by SDS-PAGE, western blots were detected with anti-mouse IgG. A representative blot is shown from 6 individual experiments. B) Neutralization of GFP+ GT1 parasites coated with heat inactivated serum ($10^1$ dilution) from mice on day 5 of GT1 secondary infection. Parasites were incubated in serum for 20 minutes before infection of mouse embryonic fibroblasts (MEFs) and assessed by FACS 2h later. The fraction of infected host cells (GFP+ cells) is normalized to that of parasite infections without serum. C) As in B, but sera-coated parasites ($10^3$ dilution) were allowed to invade A/J or C57BL/6J bone marrow derived macrophages (BMDMs) for 40 minutes. Opsonization was quantified by microscopy using a single blind analysis. Plotted is the fraction of Lamp-1+ GFP+ parasites over total GFP+ parasites that had either a GRA7+ parasitophorous vacuole and evaded opsonization or were Lamp1+ and were opsonized. For B and C, sera were obtained from multiple experiments, each dot represents an individual mouse; unpaired t-tests * P<0.05, ** P<0.01. D) Gating strategies for identifying memory B cells. Memory B cells are identified as CD19+ B220 + CD23+ CD21^mid CD73+. Conventional memory B cells are FCRL5- CD80- while atypical memory B cells are identified as FCRL5+, CD80+. Representative FACS plots

of memory compartments in A/J and C57BL/6J mice on day 5 of secondary infection with the type I GT1 strain are shown. The frequency of class-switched (IgM- IgD-) memory cells at the indicated infection states were analyzed. Each dot represents the results from an individual mouse and the cumulative averages +SD from 2 experiments are plotted. N = 3–6 mice per infection state. Significance was assessed with an unpaired two-tailed t-test; **** P<0.0001, ** P<0.01.

peritoneal B-1a, B-1b and B-2 cell response characteristics in resistant and susceptible mice during naïve, chronic and secondary infection states (S2 Dataset). GO term enrichment analysis consistently found type I and II interferon signaling and immune defense signatures as being enriched in the most differentially regulated genes among B cells following infection irrespective of genetic background (S10A Fig). Looking specifically at CD5- B-1b cells in A/J mice, which bore evidence for enhanced activation (Fig 6), pathway enrichment analysis of genes differentially upregulated on day 5 of secondary infection compared to the naïve state found additional signatures of TLR-signaling, complement activation and somatic recombination (S10A Fig). Germline transcription of *Ighg1*, *Ighg3*, *Ighg2b*, and *Ighg2a/c* was greatly enhanced in CD5- B-1b cells from A/J compared to C57BL/6J mice on day 5 of secondary infection (Fig 6F), suggesting heightened isotype class switching occurred in mice in the resistant background. Consistent with their IgM-IgD- phenotype (Fig 6B and 6C), both B-1a and B-1b cells underwent significant IgG isotype class-switching as revealed by intracellular staining, which was readily observed in the splenic environment of A/J mice (Fig 6G and 6H) where activated B-1 cells migrate to secrete antibody [57–59]. Moreover, *Scimp* an adaptor for TLR4 signaling [60], *Semaphorin7* a noted inducer of inflammatory cytokines [61], the alarmins *S100a8* and *S100a9*, and genes associated with tissue tolerance including *Retnlg* and *Slpi*, were specifically upregulated in CD5- B-1b cells from A/J compared to C57BL/6 mice on day 5 of challenge (S10B Fig). Gene Set Enrichment Analysis (GSEA) of CD5- B-1b transcriptional variation between mice on day 5 of challenge detected correlation with gene sets that distinguish B cells following vaccination with different TLR-stimulating adjuvants (MPL vs R848, GSE25677) (S10C Fig). In summary, the transcriptomic data suggest in the resistant A/J background, but not in C57BL/6J mice, B-1 cells undergo extensive IgG class switch recombination, perhaps through enhanced TLR-signaling.

## Gene dosage of *Nfkbid* impacts parasite-specific IgG1 responses

Because A/J *Nfkbid* polymorphisms are largely found in non-coding regions, including intronic and untranslated regions (UTR) (S1 Dataset), we hypothesized that B cell differences observed in our system could be due to differences in *Nfkbid* expression levels. To investigate this possibility, we quantified *Nfkbid* expression within our transcriptomic dataset as well as by qPCR. B cells from C57BL/6J mice had greater expression of *Nfkbid* relative to A/J, particularly at the chronic infection stage (Fig 7A). Because *Nfkbid* expression differences could reflect heightened responsiveness to parasite load, we stimulated enriched peritoneal and splenic B cells from A/J and C57BL/6J mice with LPS and noted that B cells from C57BL/6J mice had on average 1.5- to 2-fold greater induction of *Nfkbid* transcripts relative to A/J (Fig 7B).

To mimic the lower *Nfkbid* gene expression observed in A/J mice and explore how gene dosage of *Nfkbid* impacts antibody responses to *T. gondii*, C57BL/6J x *bumble* F1 mice were generated (*Nfkbid*+/-). Serum from chronically infected *Nfkbid*+/-, C57BL/6J, and A/J mice were analyzed. Although parasite-specific IgM did not differ between mouse strains, increased parasite-specific IgG1 responses were observed in A/J relative to C57BL/6 mice, a response that was phenocopied for IgG1 in *Nfkbid*+/- mice (Fig 7C). The increase in IgG1 was also reflected in CD138+ plasma cell differentiation in *Nfkbid*+/- mice (Fig 7D and 7E), consistent with previous reports that *Nfkbid* regulates plasma blast and IgG1 responses [38]. Hence, while *Nfkbid* is required for the generation of IgM and robust IgG responses against *T. gondii*, gene

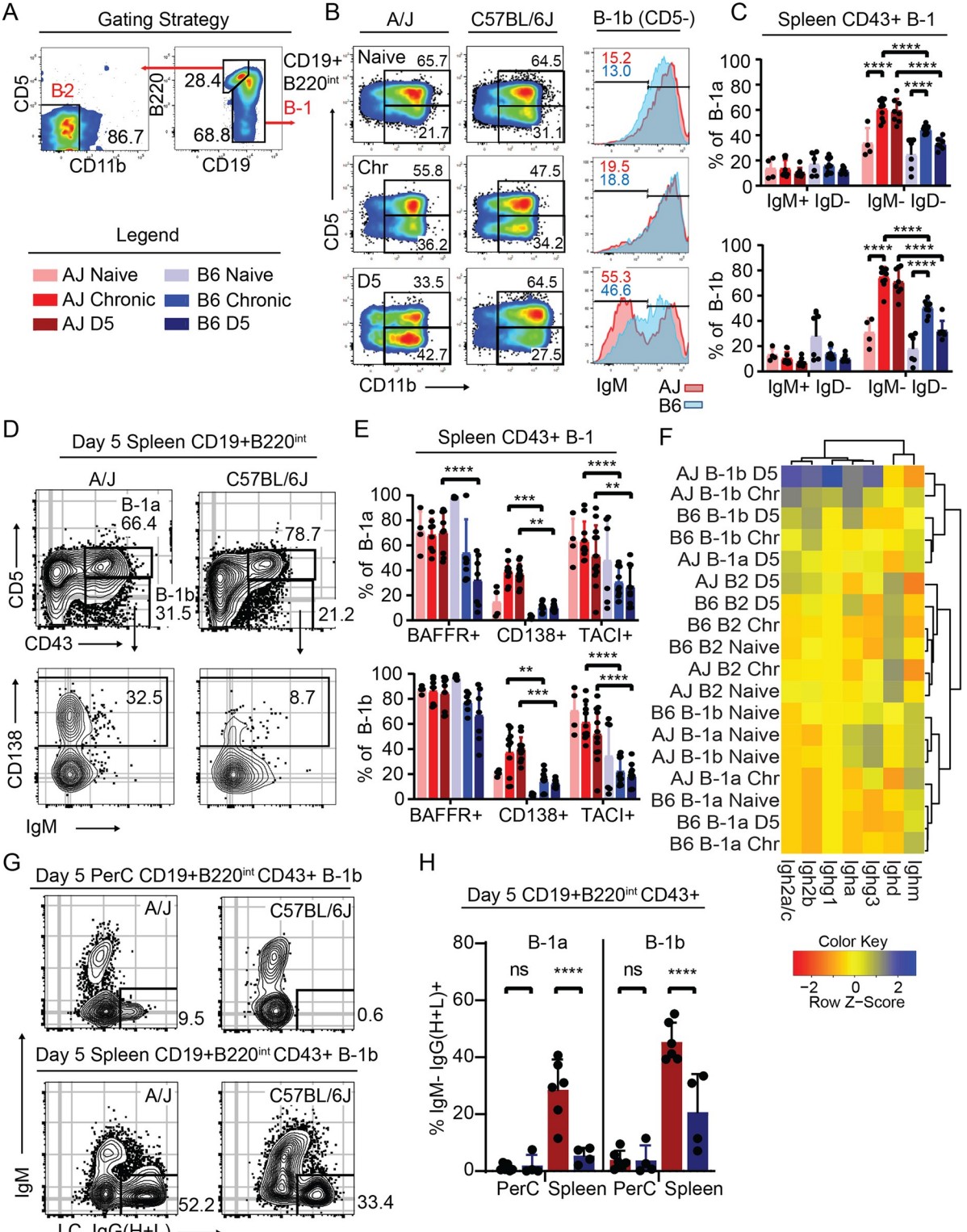

**Fig 6. Evidence for enhanced B-1 cell activation in resistant A/J mice.** A) Gating strategies for identifying B-1 (CD19+ B220^int-neg) and B-2 (CD19+ B220^hi) B cells. The legend applies to panels C-H. B) Representative FACS plots of the CD11b+ peritoneal B-1 B cell compartment in A/J and C57BL/6J (B6) mice at naïve, chronic (Chr), and 5 days (D5) post-secondary infection with the GT1 strain. Numbers indicate the percent of cells that fall within the indicated gate. Representative histograms of IgM surface expression and percent of cells that fall within the IgM^lo gate of CD5- B-1b cells from A/J (red) and C57BL/6J (blue) at the indicated time points. C) Frequencies of splenic CD43+ B-1a (CD5

+) or B-1b (CD5-) that are IgM+IgD- or IgM-IgD- in A/J and C57BL/6J mice at the indicated infection states. D) Representative FACS plots of splenic CD19+ B220^int-neg B cells stained for CD43 and CD5 in A/J and C57BL/6J mice at D5 of secondary infection. Representative CD138 expression on CD5- CD43+ B-1b cells. E) Frequencies of CD43+ splenic B-1a and B-1b cells from A/J and C57BL/6J mice that express BAFFR+, TACI+, or CD138+ at the indicated infection states. F) Heatmap depicting the relative expression of all *Ighg* transcripts from the indicated B cell population, mouse strain and infection state. G) Representative FACS plots of intracellular IgG (H+L) of B-1b cells, and H) frequency of both peritoneal and splenic B-1a and B-1b cells of A/J and C57BL/6J mice at day 5 of secondary infection. For C, E and H, the cumulative average +SD from 2–4 experiments is plotted and each dot represents the result from an individual mouse; P values calculated by 2-way ANOVA with Tukey correction; **** P<0.0001, *** P<0.001, ** P<0.01, * P<0.05.

dosage of *Nfkbid* further controls IgG1 isotype profiles, presenting *Nfkbid* as tunable modulator of antibody IgG responses to parasites (S11 Fig). Of note, when *Nfkbid*+/- mice were given secondary infections with the GT1 strain, all mice succumbed to the challenge (S3B Fig), observations that are consistent with a multiple-QTL model for *T. gondii* immunity and apparent need for additional modifiers on chromosomes 10 and 17 to survive highly virulent challenges.

## Discussion

The overarching question that shaped the experiments shown within this study is: how do we achieve protective immunity against highly virulent *T. gondii* strains? We utilized a 40-year-old recombinant inbred mouse panel [62] to answer this question. While this recombinant inbred panel had been previously used within the *T. gondii* model to answer questions regarding primary infection [30,31] and macrophage responses [32], our work presented here is the first to extend the model to find novel host factors for protective immunological memory responses. Using a forward genetic approach, we identified 4 loci that correlated with survival against secondary infection. Investigation of the first gene candidate on chromosome 7, *Nfkbid*, found a host factor that was not required for survival against primary infection but was required for survival against secondary infection, highlighting the efficacy of both our screen and the A/J versus C57BL/6J model.

Our data revealed that *Nfkbid* is a central regulator of humoral immunity to *T. gondii*, with no obvious impact on the effector T cell response in *bumble* mice. Within the C57BL/6J background, *Nfkbid* is required for maturation and class switching of B-2 cells during chronic *T. gondii* infection. While B-2 responses dominate the antibody response in this background, the lack of B-1 cells appear partially responsible for the immunity defect observed in *bumble* mice and are required for complete immunity in bone marrow chimeras. In resistant A/J mice, survival against *T. gondii* infection correlates with a strong layered humoral response: enhanced activation and class-switching in both B-1 and B-2 cells. In this context, B-1 cells may assist the B-2 response to provide full immunity to challenge. The ability of B-1 cells to make parasite-specific antibodies, though of lower affinity, potentially amplifies B-2 immune responses to *T. gondii* through internalization of B-1 cell-derived antigen-antibody complexes [63], assisting MHCII antigen presentation for CD4 T cell help [64]. However, the exact protective function of B-1 cells remains unknown within this model. The only report of B-1 mediated immunity to *T. gondii* linked increased nitric oxide and splenic cytokine levels as correlates of protection [12]. Our data suggests the B-1 mechanism of immunity in resistant A/J mice is through antibody generation, however there are known antibody-independent mechanisms for B-1 cells such as immunomodulation via B-1 derived IL-10 [65] and microbial clearance and antigen presentation via phagocytosis [66]. A closer examination of these and related functions may reveal the mechanism by which B-1 contribute to protection. Whatever the case may be, the effect of B cell mediated immunity to *T. gondii* and other pathogens is likely underestimated in murine models using the C57BL/6 background, as enhanced B-1 and B-2 responses were primarily observed in A/J mice.

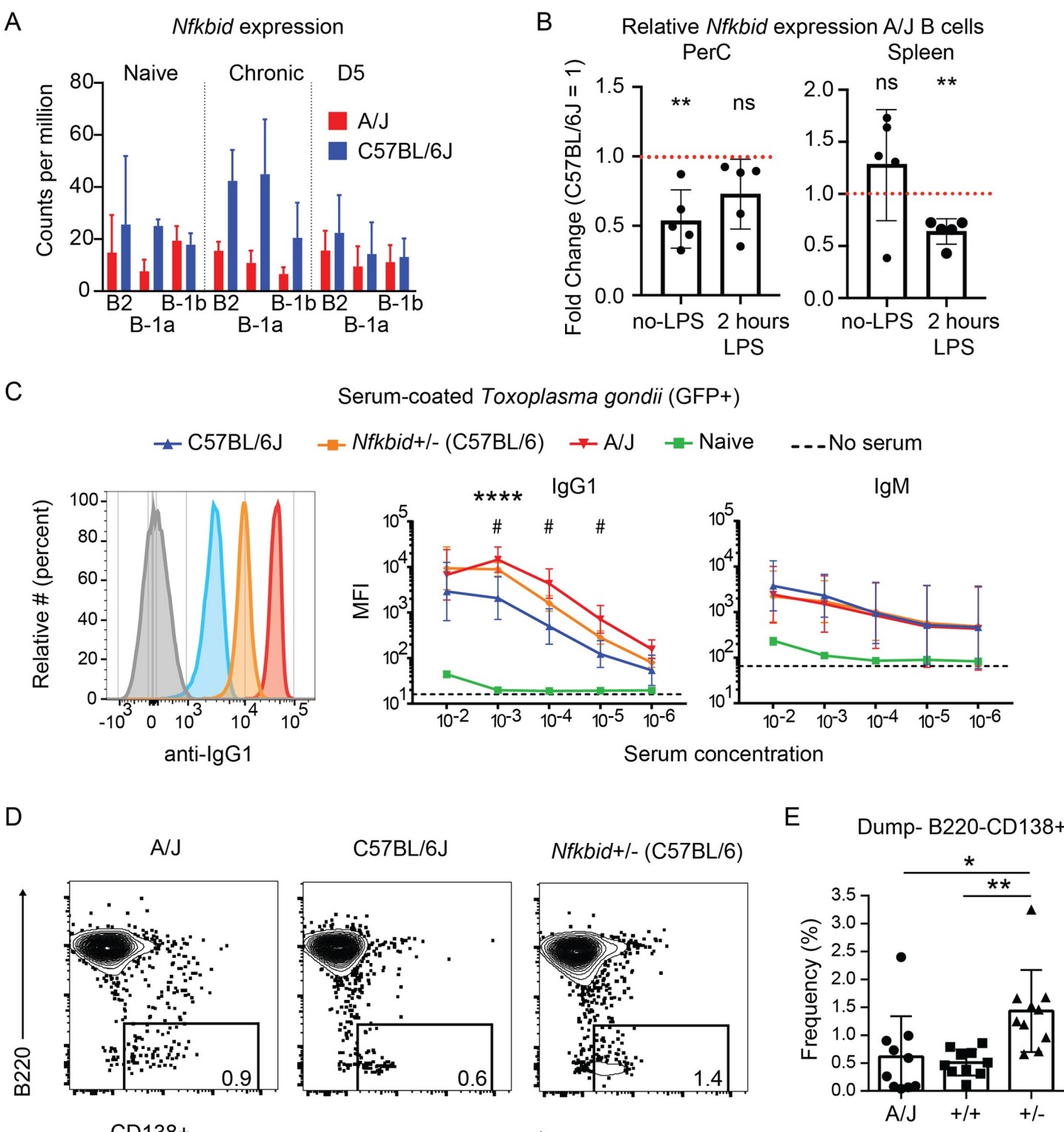

**Fig 7. Gene dosage of *Nfkbid* impacts parasite-specific IgG1 responses.** A) *Nfkbid* expression in CPM (Counts per million) of 3'-Tag RNA-seq reads of the indicated B cell populations obtained from A/J and C57BL/6J mice that were either naïve, chronically infected, or on D5 of secondary infection with the type I GT1 strain. B) Enriched B cells from the PerC and spleen were stimulated with LPS for 2 hrs and *Nfkbid* transcripts were quantified by qPCR; ** P< 0.01, paired t-test. C) Representative histograms display the detection of parasite-specific IgG1 bound to formaldehyde fixed GFP+ type I GT1 parasites; diluted serums (10³) from C57BL/6J, *Nfkbid*+/- (C57BL/6J x *bumble* F1), and A/J mice chronically infected with the type III CEP strain were assayed. Anti-mouse IgG1 background staining in the absence of serum is

shown. Also shown is the quantification of the parasite-specific IgG1 and IgM antibody isotype binding over a range of serum concentrations. Plotted is the cumulative average +/-SD of the MFI from 2–3 separate experiments (C57BL/6J n = 8; *Nfkbid*+/- n = 7; A/J n = 8); significance was assessed by unpaired t-tests and Holm-Sidak corrections comparing A/J vs C57BL/6J (*) or *Nfkbid*+/- vs C57BL/6J (#); **** P<0.0001, # P<0.05. IgG1 staining was not significantly different between A/J and *Nfkbid* +/- serums; IgM staining was not significantly different between infected mice. D) Representative FACS plots of the dump- CD19+ CD138+ plasmablast populations within A/J, C57BL/6J and *Nfkbid*+/- mice at day 5 of secondary infection with type I GT1 parasites. F) Frequency of B220- CD138+ plasmablasts of total live dump- CD19 + cells on day 5 of secondary infection. Plotted is the cumulative average +/-SD of 2 separate experiments (n = 10 per mouse strain); significance was assessed by unpaired t-tests; * P< 0.05.

Though the signaling pathway remains to be investigated, *Nfkbid* is downstream of TLR signaling [34,35,37], which in B-1a cells causes them to downregulate CD5 and facilitate differentiation into antibody secreting cells [53,67]. CD5 is a potent negative regulator of antigen receptor signaling that renders B-1a cells unresponsive to B cell receptor (BCR)-triggering. This inhibition is overcome by TLR-stimulation which causes CD5 to dissociate with the BCR, thereby releasing repression of BCR-mediated signaling and antibody secretion [67] against foreign- and self-antigen [53]. These data suggest CD5- B-1b cells may represent an activated state of B-1 cells, and call into question a strict division of labor between these two subsets. This supposition would fit several of the observations made in our system, including evidence for enhanced class switch recombination and TLR-gene signatures observed in the CD5- B-1b cells of the resistant background. In addition, BAFFR and TACI are known inducers of class-switch recombination [68], and increased expression of these receptors may further lower the threshold of activation and differentiation into CD138+ plasmablasts/plasma cells, all of which are regulated by *Nfkbid* [37] and occurring with greater magnitude in B-1 cells of genetically resistant A/J mice. Although class switch recombination occurs with much greater frequency in B-2 cells of A/J mice, we found no evidence for enhanced expression of BAFFR and TACI in this compartment following *T. gondii* infection. Further investigation of pathways upstream of *Nfkbid* has the potential to elucidate key requirements for *T. gondii* immunity.

*Nfkbid* appears to regulate transitional development in B-2 cells, which is analogous to previous findings in B-1 cells [50], but only evident following *T. gondii* infection. As immature B cells migrate out of the bone marrow to the spleen, there are several checkpoints which are controlled by NF-κB such as BCR- or BAFF-mediated signaling [69], both of which are regulated by *Nfkbid* [37]. Our observation of an accumulation of transitional B cells during *T. gondii* infection in *bumble* mice suggests *Nfkbid* plays a role in stabilizing advancement out of these developmental checkpoints. *Nfkbid* could act as a negative regulator of BCR signaling, enabling pathogen-reactive B-2 cells to develop beyond negative selection that would otherwise occur as antigen accumulates in secondary lymphoid organs over time. Alternatively, *Nfkbid* could be a positive regulator of NF-κB signaling, increasing the strength of BCR signaling to enable transitional B cells to become mature B cells. In both cases, a developmental defect likely restricts the pool of *T. gondii*-reactive mature B cells, preventing replenishment of antibody secreting cells during infection, culminating in the low parasite-specific antibody titers observed in *bumble* mice.

Though *Nfkbid* has a large effect on humoral responses, our study does not address whether this is due to a B cell intrinsic or extrinsic effect. Moreover, we found evidence that non-hematopoietic cells contribute to *Nfkbid*-dependent immunity (Fig 4E), indicating its role in multiple compartments. Previous studies using model T-dependent antigens, such as 2,4,6, Trinitrophenyl (TNP)-KLH and 4-hydroxy-3-nitrophenylacetyl (NP)-ovalbumin (OVA), have demonstrated the *Nfkbid*-dependent IgM response is B cell intrinsic [34], but the IgG1 response is not and requires *Nfkbid*-dependent CD4 T cell help [38]. Additionally, C57BL/6J mice undergo T cell exhaustion during infection with *T. gondii* [27,47], and during *Plasmodium* infection, PD-L1 and LAG3 blockade improves germinal center and plasma cell

responses via an enhanced T follicular help [70]. T cell exhaustion may serve as an explanation for why C57BL/6J have both reduced humoral responses and IFNγ+ CD8 T cells relative to A/J mice (S9 Fig). Whether the humoral or IFNγ differences were due to a T cell intrinsic phenotype regulated by *Nfkbid*, or overall T cell dysfunction induced by high virulence strains in the C57BL/6J background [27], was not explored in this study. Investigation of the major cell types driving *Nfkbid*-dependent humoral responses has the potential to elucidate key requirements for *T. gondii* immunity.

It is important to emphasize that multiple polymorphisms determine the complex phenotype of secondary infection immunity to *T. gondii*. Our genetic screen revealed at least 4 loci that each account for 20–40% of the overall heterologous immunity to *T. gondii*, and that the H-2 locus can be an important modifier of resistance against certain parasite strains. Perhaps not surprisingly, the *Nfkbid* polymorphism in a stand-alone fashion did not fully restore immunity, as inferred from chromosome 7 consomic mice and by our attempts to mimic the lower gene expression observed in resistant mice through heterozygous expression. Whereas polymorphic *Nfkbid* contributes 24% to this phenotype, perhaps by regulating plasma cell differentiation (Fig 7), this smaller effect QTL was instrumental in identifying *Nfkbid*, where a more drastic gene inactivation revealed its role in multiple compartments in *bumble* mice, notwithstanding its requirement for humoral immunity.

In summary, heterologous immunity to a parasitic pathogen should, at a minimum, prevent disease against a wide variety of strains that differ in virulence or polymorphic antigens. An ideal parasite vaccine would entirely protect against re-infection and induce sterile immunity, thought possible since re-infection studies were first performed in mice [71] and humans immunized with irradiated sporozoites and *Plasmodium sp.* challenge [72]. Yet, only one partially protective vaccine is in use for any human parasitic pathogen, RTS,S/AS01, which has low efficacy for malaria prevention [73]. Our findings highlight the role of both innate and conventional B cells in humoral immunity to *T. gondii*, introducing B-1 cells as a potential vaccine target along with B-2 cells to maximize humoral immunity to parasitic infections. Moreover, we present a modulator of antibody responses against parasitic infections, *Nfkbid*, a transcriptional regulator that can tune B cell responses to provide an overall effective class-switched antibody response against parasites.

## Material and methods

### Ethics statement

Mouse work was performed in accordance with the National Institutes of Health Guide to the Care and Use of Laboratory Animals. All protocols have been reviewed and approved by UC Merced's Committee on Institutional Animal Care and Use Committee (protocol AUP 20–0015). UC Merced has an Animal Welfare Assurance filed with OLAW (#A4561-01), is registered with USDA (93-R-0518), and the UC Merced Animal Care Program is AAALAC accredited (001318). Every effort was made to ensure unnecessary stress was placed on the animals.

### Mice

Female C57BL/6J (H-2b), A/J (H-2a), C57BL/10SnJ (H-2b), B10.A-*H2ᵃ H2-T18ᵃ*/SgSnJ (H-2a), B6AF1/J (A/J x C57BL/6J F1 progeny), B6.129S2-*Ighm^{tm1Cgn}*/J (muMT), B6.SJL-*Ptprcᵃ Pepcᵇ*/BoyJ (CD45.1 congenic), B6.Cg-*Gpi1ᵃ Thy1ᵃ Ighᵃ*/J (IgH-a triple congenic mice), C57BL/6J-Chr7^{A/J}/NaJ and C57BL/6J-Chr10^{A/J}/NaJ (chromosome 7 and 10 consomic mice), and 26 (AxB;BxA) recombinant inbred (RI) mice derived from A/J and C57BL/6 founders, were purchased from Jackson Laboratories. The *bumble* mouse line used for this research project [35], C57BL/6J-*Nfkbid*^{m1Btlr}/Mmmh, RRID:MMRRC_036725-MU, was obtained from the

Mutant Mouse Resource and Research Center (MMRRC) at University of Missouri, an NIH-funded strain repository, and was donated to the MMRRC by Bruce Beutler, M.D., University of Texas Southwestern Medical Center. *Bumble* mice were crossed to C57BL/6J to generate F1 *bumble* heterozygotes (*Nfkbid*+/-). Mice were maintained under specific pathogen free conditions at UC Merced.

## Parasite strains and cell lines

Human foreskin fibroblasts (HFFs) monolayers were grown in DMEM (4.5 g/L D-glucose) (Life Technologies) supplemented with 2 mM L-glutamine, 20% fetal bovine serum (FBS) (Omega Scientific), 1% penicillin-streptomycin, and 0.2% gentamycin (Life Technologies). Mouse Embryonic Fibroblasts (MEFs) were grown in DMEM (4.5 g/L D-glucose) (Life Technologies) supplemented with 10% fetal bovine serum (FBS) (Omega Scientific), 20mM HEPES, 1% penicillin-streptomycin, and 0.2% gentamycin (Life Technologies). *Toxoplasma gondii* strains were passaged in HFFs in 'Toxo medium' (4.5 g/L D-glucose, L-glutamine in DMEM supplemented with 1% FBS and 1% penicillin-streptomycin). The following clonal strains were used (clonal types are indicated in parentheses): RH *Δku80 Δhxgprt* (type I), RH (1–1) *GFP::cLUC* (type I), GT1 (type I), GT1 *GFP::cLuc* (type I), and CEP *hxgprt-* (type III). The following atypical strains were used: MAS, MAS *GFP::cLuc* (2C8) (haplogroup 'HG' HG4), GUY-MAT (HG5), FOU (HG6), GPHT (HG6), TgCATBr5 (HG7), GUY-DOS (HG10), and VAND (HG10). The uracil auxotroph vaccine strain, RH *Δup Δompdc* [74], was passaged in HFFs in medium containing 250μM uracil.

## Generation of GFP-expressing GT1 strains

GT1 parasites were transfected with linearized plasmids for parasite expression of GFP and click beetle luciferase (*GFP::cLUC*), parasites were grown on HFF monolayers in T-25 flasks in Toxo medium for 2 weeks. Parasites were removed from the flasks by scraping; the parasites were pelleted and washed with PBS and suspended in sterile FACS buffer (2% FBS in PBS). Fluorescent parasites where then sorted via fluorescence-activated cell sorting (FACS) into a 96-well plate with confluent HFF monolayers. To ensure single plaque formation in at least one of the wells, the sort was titrated using the following parasite numbers: 100, 50, 25, 12, 6, 3, 2, and 1 for each well per row of 8.

## Parasite infections

Parasite injections were prepared by scraping T-25 flasks containing vacuolated HFFs and sequential syringe lysis first through a 25G needed followed by a 27G needle. The parasites were spun at 400 rpm for 5 min to remove debris and the supernatant was transferred, followed by a spin at 1700 rpm washing with PBS. For primary infections, mice were infected intraperitoneally (i.p.) with $10^4$ tachyzoites of type III CEP *hxgprt-*. For some experiments, mice were vaccinated i.p. with $10^6$ tachyzoites of RH *Δup Δompdc*. For secondary infections, mice were infected i.p. with $5x10^5$ type I parasites (RH or GT1). Parasite viability of the inoculum was determined by plaque assay following i.p. infections. In brief, 100 or 300 tachyzoites were plated in HFF monolayers grown in a 24-well plate and 4–6 days later were counted by microscopy (4x objective).

## Blood plasma isolation and assessment of seroconversion

All mice were assessed for sero-positivity to *T. gondii* 4–5 weeks post primary infection. 50μL of blood was isolated from mice in tubes containing 5μL of 0.5M EDTA on ice, pelleted and

the supernatant containing blood plasma was heat-inactivated to denature complement at 56˚C for 20 minutes and then stored at -80˚C. HFFs were grown on coverslips and infected with GFP-expressing RH (1–1) overnight, fixed 18 hrs later with 3% formaldehyde (Polysciences) in PBS, washed, permeabilized and blocked with PBS containing 3% bovine serum albumin Fraction V (Sigma), 0.2M Triton X-100, 0.01% sodium azide, incubated with a 1:100 dilution of collected blood plasma for 2 hrs at room temperature, washed with PBS, and detected with Alexa Fluor 594-labeled secondary antibodies specific for mouse IgG (cat # A11032, Life Technologies). Seropositive parasites were observed by immunofluorescence microscopy (Nikon Eclipse Ti-U).

### Brain superinfection assays and cyst enumeration

Brains from chronically infected mice (CEP *hxgprt-*) that survived secondary challenge were dissected, rinsed in PBS, passed through a 21G needle several times, pelleted and suspended in 1mL of PBS. For rederivation, 100μL of the brain homogenate was used to inoculate HFF monolayers in Toxo medium. One to two weeks later, infected HFFs were syringe-lysed and plated on new HFF monolayers to encourage parasite growth. Once HFFs were fully vacuolated, parasites were passaged in Toxo medium supplemented with mycophenolic acid (MPA) and xanthine that selects for parasites encoding a functional *HXGPRT* (i.e. the challenging strains) and against the chronically infecting type III *hxgprt-* which lacks a functional *HXGPRT* gene. Outgrowth in MPA-xanthine was considered evidence for superinfection.

For counting tissue cysts, 100μl of brain homogenate was fixed in 900μL of ice cold methanol, incubated for 5 minutes in microtubes (MCT-175-C, Axygen), washed and stained overnight in a 500μL PBS solution containing 1:150 dilution of FITC-conjugated *Dolichos biflorus* agglutinin (Vector Laboratories) with slow rotation at 4˚C. The stained homogenate was further washed and suspended in 1mL of PBS, of which several 50μL aliquots were counted by fluorescence microscopy, and the number of cysts per brain were deduced.

### Genetic linkage analysis

Quantitative trait loci (QTL) analysis was performed with the package r/QTL in R (version 3.6.1). LOD scores for each marker were calculated using the Haley-Knott regression model with the function 'scanone', or for all possible combination of two markers (i.e. epistatic interactions) using the function 'scantwo'. 1000 permutations were performed to obtain the genome wide LOD threshold for a P value of ≤0.05, which was considered statistically significant. Similar results were obtained with a linear mixed regression model. To estimate the effect each QTL had on the overall phenotype, the function 'fitqtl' was first used to fit the data to a multiple-QTL model. Statistical support was found for inclusion of all four QTLs with LOD scores > 3 compared to any lesser combination of three-QTLs (ANOVA P <0.02). Individual QTL effects were then calculated under the assumption of the four-QTL model, which collectively accounts for 91% of the observed phenotypic variance. No statistical support was found for epistatic interactions between QTLs.

### Cell isolation, in *vitro* recall infections, and FACS analysis

PECs were isolated by peritoneal lavage and splenocytes obtained, as described in [27]. In brief, 4mL of FACS buffer (PBS with 1% FBS) and 3mL of air were injected into the peritoneal cavity with a 27G needle. After agitation, the PEC wash was poured into a conical tube. PEC washes were filtered through a 70μm cell strainer, pelleted, and washed with FACS buffer before staining. Spleens were dissected and crushed through 70μm cell strainers, pelleted, incubated in ACK red blood cell RBC lysis buffer (0.15M NH₄Cl, 10mM KHCO₃, 0.1mM EDTA)

for 5 minutes at room temperature, then washed with FACS buffer. To obtain peripheral blood leukocytes (PBLs), 50μL of blood was isolated from mice in tubes containing 5μL of 0.5 M EDTA on ice, pelleted and incubated in ACK lysis buffer, washed and peripheral blood leukocytes (PBLs) were suspended in FACS buffer.

For FACS analysis, all preparations were done on ice, and cells were blocked in FACS buffer containing Fc Block anti-CD16/32 (2.4G2) (BD Biosciences), 5% normal hamster serum, and 5% normal rat serum (Jackson ImmunoResearch) for 20 minutes prior to staining with fluorophore-conjugated monoclonal antibodies (mAbs). The following mAbs (1:100 staining dilutions) were used: anti-CD1d-BV650 (1B1, BD Bioscience), anti-CD11c-eFlour 450 (N418, eBioscience); anti-CD11c-eFlour 450 (N418, BD Bioscience); anti-CD45.2-eFlour 450 (104, eBioscience); anti-CD4-eFlour 450 (GK1.5, eBioscience), anti-CD4-PECy7 (GK1.5, eBioscience); anti-CD11b-FITC (M1/70, eBioScience), anti-CD11b-BUV395 (M1/70, BD Bioscience), anti-CD11b-BV421 (M1/70, BD Bioscience), anti-CD11b-Pacific Blue (M1/70, BioLegend); anti-IFNγ-PE (XMG1.2, BD Bioscience); anti-CD8α-APC (53–6.7, eBioscience), anti-CD8α-BV510 (53–6.7, BioLegend), anti-Ly6G-APC (1A8-Ly6g, eBioscience), anti-CD19-PerCP-Cy5.5 (ebio1D3, eBioscience), anti-CD19-PE (6D5, BioLegend), anti-CD19-BV785 (6D5, BioLegend), anti-CD3-eFlour 780 (17A2, BD Biosciences), anti-Ly6C--PECy7 (HK1.4, BioLegend), anti-CD23-Pacific Blue (B3B4, BioLegend), anti-CD23-AF700 (B3B4, BioLegend), anti-CD21/CD35-FITC (7E9, BioLegend), anti-CD21/CD35-PE (7E9, BioLegend), anti-CD5-APC (53–7.3, BioLegend), anti-CD43-BV510 (S7, BD Bioscience), anti-CD43-BUV737 (S7, BD Bioscience), anti-CD5-PerCP-Cy5.5 (53–7.3, BioLegend), anti-CD5-Cy7-APC (53–7.3, BioLegend), anti-CD45R/B220-Cy7-APC (RA3-6B2, anti-CD45R/B220-BUV-661 (RA3-6B2, BD Bioscience), anti-CD73-Cy7-PE (eBioTY/11.9, eBioscience), anti-CD80-BV711 (16-10A1, BioLegend), anti-FCRL5-af88 (polyclonal, R&D systems) anti-IgM-PECy7 (RMM-1, BioLegend), anti-IgM-BV605 (RMM-1, BioLegend), anti-IgD-FITC (11-26c.2a, BioLegend), anti-IgD-PEDazzle (11-26c.2a, BioLegend), anti-CD138/Syndecan-1-BV510 (281–2, BD Bioscience), anti-CD138/Syndecan-1-BV650 (281–2, BioLegend), anti-mouse-CD267/TACI-AlexaFlour-647 (8F10, BD Biosciences), and anti-CD268/BAFF-R-PE (7H22-E16, BioLegend). Other FACS reagents include the viability dye propidium iodide (Sigma) used at a final concentration of 1μg/mL.

For *in vitro* recall, splenocytes and PerC were isolated from chronic and challenged mice (day 5 or 7 following secondary infection) and $6x10^5$ cells per well (96-well plate) were plated in T cell medium (RPMI 1640 with GlutaMAX, 20% FBS, 1% Pen/Strep, 1mM NaPyruvate, 10mM HEPES, 1.75μl BME). Cells were infected with a type I strain (RH or GT1) strain at an MOI (multiplicity of infection) of 0.2 for 18 hr; 3μg/mL brefeldin A (eBioscience) was added for the last 5 hr of infection. 96-well plates were placed on ice, cells were harvested by pipetting and washed with FACS buffer, blocked, and stained for surface markers. Cells were fixed with BD Cytofix/Cytoperm and permeabilized with BD Perm/Wash solution (cat# 554714, BD Pharmingen), stained with anti-IFNγ-PE (XMG1.2, BD Bioscience), anti-GZB (GB11, BioLegend) and anti-IL-2-APC (JES6-5H4, BioLegend) on ice for 1 hr or overnight. Cells were then washed once with BD Perm/Wash solution, once in FACS buffer, and analyzed by FACS.

For FoxP3 staining, peritoneal lavage was performed on chronic and challenged (day 7 following secondary infection) *bumble* and WT mice. Cells were washed and surface stained. Fixation and permeabilization was performed with the eBioscience Foxp3/Transcription Factor Staining Buffer Set (cat # 00-5523-00) before intracellular staining with anti-Foxp3-PE (MF-14, BioLegend) according to the manufacturer's recommendations. Flow cytometry was performed on a Beckman Coulter Cytoflex LX, LSR II (BD Biosciences), or the Bio-Rad ZE5 and analyzed with FlowJo software.

For intracellular staining of IgG H/L, PerC and splenocytes were surface stained with anti-CD43-BV510 (S7, BD Bioscience), anti-CD19-PE (6D5, BioLegend), anti-IgD-FITC (11-26c.2a, Biolegend), anti-IgM-PE/Cy7 (RMM-1, BioLegend), anti-CD5-APC (53–7.3, BioLegend), and anti-CD45R/B220-Cy7-APC (RA3-6B2). Cells were then fixed with BD Cytofix/Cytoperm and permeabilized with BD Perm/Wash solution (cat# 554714, BD Pharmingen), stained with anti-IgG(H+L)-a350 (Polyclonal, Invitrogen) for 30 minutes. Cells were then washed once with BD Perm/Wash solution, once in FACS buffer, and analyzed by FACS.

## Analysis of isotype-specific antibody reactivity to *Toxoplasma gondii* by flow cytometry

For serum reactivity analysis, syringe-lysed GFP-expressing strains (RH1-1 and GT1-GFP) were fixed in 3% formaldehyde for 20 minutes, washed twice in PBS, and plated in 96 well micro-titer plates at $4x10^5$ parasites/well. The parasites were then incubated with serum from chronically infected mice, at serum concentrations ranging from $10^{-2}$ to $10^{-6}$ diluted in FACS buffer, for 20 minutes at 37˚C. Parasites were then washed with FACS buffer and placed on ice for incubation with anti-isotype detection antibodies depending on the application: anti-IgG3-BV421 (R40-82, BD Bioscience), anti-IgM-PE/Cy7 (RMM-1, BioLegend), anti-IgG1-FITC (RMG1-1, BioLegend), anti-IgG1-APC (RMG1-1, BioLegend), anti-IgG2b-FITC (RMG2b-1, BioLegend), anti-IgG2b-PE (RMG2b-1, BioLegend), anti-IgG2a-FITC (RMG2a-62, BioLegend), anti-IgG2a-PerCP/Cy5.5 (RMG2a-62, BioLegend), anti-IgM-a-PE (DS-1, BD Bioscience), and anti-IgM-b-PE (AF6-78, BD Bioscience).

## Parasite neutralization assay

Heat-inactivated blood plasma, 56˚C 30 minutes, was used to coat live GFP+ parasites for 20 minutes at 37˚C before infecting $5x10^5$ mouse embryonic fibroblasts/well in 96 well plates. Immediately following addition of parasite to MEF wells, plates were spun at 1200rpm for 3 minutes to synchronize infection. 2 hrs after initiation of infection, cells were placed on ice and harvested by scraping with pipette tips. Cells were washed twice in FACS buffer, suspended in 1:1000 PI in FACS buffer, and then analyzed by flow cytometry.

## Parasite opsonization assay

Bone marrow derived macrophages (BMDMs) were generated as previously described [26], and plated on coverslips in 24 well plates in medium (4.5 g/liter D-glucose DMEM with Gluta-MAX (Gibco), 10% heat-inactivated FBS, 1% penicillin-streptomycin (Gibco), 1X non-essential amino acids (Gibco, cat# 11140076), 1mM sodium pyruvate (Gibco, cat# 11360070)) supplemented with 10% L929 conditioned medium and incubated overnight 37˚C, 5% $CO_2$. Heat-inactivated blood plasma was used to coat live GFP+ parasites for 20 minutes at 37˚C before infecting at a multiplicity of infection of 0.2. Plates were spun 1200rmp to synchronize infection and incubated for 40 minutes, 37˚C. Cells were washed and fixed with 3% formaldehyde (Polysciences cat# 18814–20) in PBS for 20 minutes, blocked and permeabilized with blocking buffer (3% BSA (SIGMA Fraction V), 5% normal goat serum (Omega), 0.2% Triton X-100, 0.1% sodium azide in PBS). To visualize the phagosomal marker Lamp-1, samples were stained with rat anti-CD107a primary antibody (BD Pharmingen, clone 1D4B) (1:500 dilution), followed by Alexa Fluor 594 goat anti-rat IgG (Life Technologies cat# A11007) secondary antibody (1:1000 dilution); GRA7 was stained with a polyclonal rabbit anti-GRA7 (1:700 dilution) (gift from John Boothroyd, Stanford University) and detected with Alexa Fluor 648 goat anti-rabbit (Life Technologies cat# A21245) secondary antibody (1:3000 dilution); DNA was stained with DAPI (Thermo Fisher, cat# 62248) (1:1000 dilution). Fluorescence

microscopy was performed (Nikon TI-S) and images were captured at 63x (CFI Plan Apochromat λ 60x; Zyla-5.5-USB3) then blinded. Images were processed (Nikon Elements) and 80–100 GFP+ events were quantified for parasite association with Lamp1+ or for containment within a ring-like GRA7+ parasitophorous vacuole (PV); fraction opsonization was calculated as = (Lamp1+GFP+ counts / Lamp1+GFP+ plus GRA7 PV+ GFP+ total counts).

## SDS-PAGE and immunoblotting for parasite lysate antigen

To generate parasite lysate antigens, *Toxoplasma gondii* was cultured in HFF and expanded to approximately $2x10^8$ parasites. Parasites were syringe-lysed, washed with sterile 1X PBS and the parasite pellet was lysed with (1mL) 0.1% TritonX-100 detergent in 1X PBS. Solubilized parasites were centrifuged at 2,000 RCF for 20 minutes to remove large debris. The supernatant was aliquoted and stored at -80˚C. Parasite lysate was reduced with β-mercaptoethanol (BME) and separated via SDS-PAGE in 4–20% Mini-PROTEAN TGX pre-cast gels (cat # 4561096, Bio-Rad) before transfer to PVDF membrane using a Trans-Blot Turbo Mini PVDF Transfer Pack (cat # 1704156, Bio-Rad) via Bio-Rad Transblot Turbo (cat # 1704150, Bio-Rad). Membranes were blocked with 10% fortified bovine milk dissolved in Tris-Buffered Saline with 0.1% Tween (TBS-T 0.1%) for 1–2 hrs at room temperature or overnight at 4˚C. Blots were then probed with heat-inactivated serum in block at either 1:1,000 dilution for serum IgM analysis or 1:5,000 dilution for serum IgG analysis overnight at 4˚C. Membranes were washed with TBS-T 0.1% three times for 20 minutes per wash. Blots were then incubated for one hr at room temperature with goat α-mouse horseradish peroxidase (HRP)-conjugated antibodies (SouthernBiotec): anti-IgM secondary 1:1000 (cat# 1020–05) and total anti-IgG secondary 1:5000 (cat# 1030–05). Membranes were then washed with TBS-T 0.1% three times and developed with Immobilon Forte Western HRP Substrate (WBLUF0500). All blots were imaged via chemiluminescence on a ChemiDoc Touch (cat# 12003153, Bio-Rad). Image Lab 6.1 software (Bio-Rad) was used for analysis of bands and total lane signal. Western blots comparing A/J to C57BL/6J were developed simultaneously.

## RNA isolation and sequencing

Peritoneal B-1a (B220$^{int-neg}$ CD19$^{high}$ CD11b+ CD5+ PI-), B-1b (B220$^{int-neg}$ CD19$^{high}$ CD11b + CD5- PI-), or B-2 B cells (B220$^{high}$ CD19+ CD11b- PI-) were sorted into 500ul RNeasy lysis buffer using a FACS ARIA II cell sorter (BD Biosciences). RNA was purified using the RNeasy mini kit (cat# 74134, Qiagen) according to the manufacturer's protocol. RNA purity was tested by Qubit (ThermoFisher) and Agilent 2100 BioAnalyzer for total RNA with picogram sensitivity. DNA libraries were generated with a Lexogen QuantSeq-UMI FWD 3′ mRNA-Seq Library Prep Kit (cat# 015). Samples were sent to UC Davis for QuantSeq 3' mRNA FWD-UMI sequencing.

## Gene expression analysis and data availability

Raw reads were trimmed and mapped by the BlueBee genomic pipeline FWD-UMI Mouse (GRCm38) Lexogen Quantseq 2.6.1 (Lexogen). In brief, reads were quality controlled with 'FASTQC', trimmed with 'Bbduk' to remove low quality tails, poly(A)read-through and adapter contaminations, read alignments to the *Mus musculus* genome build GRCm38 were done with 'STAR', and gene reads were quantified with 'HTSeq-count'. Differentially expressed gene (DEG) analysis was performed utilizing limma-voom in R version 3.6.3 in RStudio with Bioconductor suite of packages. Heatmaps were generated with 'gplots'. Pathway and GO term analysis was performed with MouseMine (mousemine.org), and gene set enrichment analysis was performed with GSEA v4.0.3.

## LPS-stimulation of enriched B cells and quantitative PCR

PerC and splenocytes were isolated from naïve 6–8 week-old C57BL/6J and A/J mice and enriched for B cells using the EasySep Mouse Pan B cell Isolation Kit (cat#19844, StemTech). Bead enrichment for splenic samples had the addition of biotinylated anti-CD43 antibodies (clone S7, BD Bioscience) to remove B-1 cells. Enriched samples were plated in a 96-well plate at 400,000 cells per well and stimulated with 25ug/ml of LPS (cat# L4391-1MG, Sigma). After 2 hrs, total RNA was isolated using the Rneasy Mini Kit (cat# 74134) and cDNA was synthesized using High Capacity cDNA Reverse Transcription Kit (ThermoFisher, cat#4368814). Quantitative PCR was performed on synthesized cDNA samples using ThermoFisher TaqMan Master Mix (cat# 4444556) and TaqMan probes: *Actb*—Assay ID:Mm02619580_g1 (cat# 4331182), and *Nfkbid*—Assay ID: Mm00549082_m1 (cat# 4331182), according to the manufacturer's protocol. Normalization of *Nfkbid* expression in each sample was calculated in comparison to *Actb* expression levels. Fold change in *Nfkbid* expression of AJ relative to that of C57BL/6J cells was determined through the delta delta CT method ($2^{-\Delta\Delta CT}$).

## PerC adoptive transfers and IgH allotype chimeric mouse generation

PerC was harvested by peritoneal lavage of 6–12 week old C57BL/6J donor mice as described above and $5\times10^6$ total peritoneal exudate cells (total PerC)/60ul PBS dose were transferred i.p. into 2–4 day old *bumble* neonates. Allotype chimeric mouse generation was performed as previously described [54]. In brief, 1-day old C57BL/6J neonates were treated with 0.1mg of anti-IgM-b (clone AF6-78) and twice weekly thereafter treated with 0.2mg of anti-IgM-b for 6 weeks. On day 2 after birth the neonates were given $5\times10^6$ total PerC from B6.Cg-*Gpi1ᵃ Thy1ᵃ Ighᵃ* delivered i.p. The mice were then allowed to rest for 6 weeks after the last antibody treatment before infection with *T. gondii*.

## Bone marrow chimeric mice generation

B6.SJL-*Ptprcᵃ Pepcᵇ*/BoyJ, *bumble*, and C57BL/6J recipient mice were given 2 doses of 500cGy with an X-Rad320 (Precision X-Ray) with a 4 hr interval. Donor BM cells were harvested from *bumble* and WT C57BL/6J mice, filtered with a 70um filter, incubated in ACK red blood cell lysis buffer, washed with PBS and transplanted by retro-orbital injection at a concentration of $10^7$ cells/200ul PBS dose. For BM chimeras reconstituted with PerC, $5\times10^6$ total PerC from WT C57BL/6J mice were transferred i.p. Recipient mice were then allowed to rest for 8 weeks. Reconstitution was assessed at 8 weeks by FACS analysis of PBL.

## ELISA

High-affinity protein binding microplates (Corning) were coated overnight goat anti-mouse IgM (1mg/ml, Southern Biotech, cat# 1021–01) and blocked with coating buffer containing with 1% BSA (w/v) and 2% goat serum (Omega Scientific) in PBS. Wells were washed with ELISA wash buffer (1X PBS, 0.05% Tween-20). Mouse serum samples were diluted 1:400 in coating buffer and incubated in the wells for 1 hr. Wells were washed 5 times and secondary goat anti-IgM-HRP (Southern Biotech, cat# 1021–08) at 1:5000 dilution was incubated for 1 hr in the wells. Wells were washed several times and developed with TMB substrate (Invitrogen). Development was stopped after 20 minutes with 1M $H_3PO_4$ stop solution. Absorbance was measured at 450nm on a BioTek Epoch microplate spectrophotometer.

## Statistics

Statistical analysis was performed with Graphpad Prism 8 software. Statistical significance was defined as P < 0.05. P values were calculated using paired or unpaired two-tailed t-tests, 2-way ANOVA with Tukey multiple comparison correction, and multiple t-tests with the Holm-Sidak correction for multiple comparisons. For non-parametric data, P values were calculated with an unpaired Mann-Whitney test. Survival curve significance was calculated using log-rank (Mantel-Cox) testing. Significance in time to death was calculated using the Gehan-Breslow-Wilcoxon test. Differential gene expression analysis statistics were calculated using the Limma-Voom R package, P values were adjusted with the Benjamini-Hochberg correction for multiple comparisons. GO term and pathway enrichment analysis statistics were performed at mousemine.org using the Holm-Bonferroni test correction. Statistical methods used for each figure are indicated in the figure legends.

## Supporting information

**S1 Fig. Naïve A/J mice are susceptible to primary infection with atypical strains.** Naïve A/J mice (n = 1) were injected i.p. with the $5\text{x}10^4$ tachyzoites of the indicated atypical strains. Survival curves are shown.
(TIF)

**S2 Fig. Cyst burden and weight during chronic infection with the low virulent type III CEP strain correlates with the murine H-2 locus.** A/J, C57BL/6J (B6), C57BL/10J (B10), and C57BL/10.AJ (B10.A) mice were injected with the type III strain *CEP hxgprt-* and allowed to progress to chronic infection. Plotted (+/- SEM) is the average cyst number (x 1000) in the brain vs. the average fraction of initial weight, where 1 is the normalized weight on the day of type III injection; the regression value ($R^2$) is indicated. Results are from 3 to 5 mice for cyst numbers (day 42 of chronic infection), and 5–12 mice for weight measurements (day 30 of chronic infection) per mouse strain.
(TIF)

**S3 Fig. Consomic and *Nfkbid*+/- C57BL/6 mice succumb to secondary infection with the type I GT1 strain.** A) Consomic mice of the C57BL/6J background with A/J chromosomal substitutions for chromosome 7 (C57BL/6J-Chr7$^{A/J}$/NaJ) or chromosome 10 (C57BL/6J-Chr10$^{A/J}$/NaJ) were infected with the type III CEP *hxgprt-* strain and allowed to progress to chronic infection. Mice were then given a secondary infection with the type I GT1 strain. B) Survival of *Nfkbid*+/- (C57BL/6J x *bumble* F1) and C57BL/6J mice against type I GT1 secondary infection. For A and B, cumulative survival is shown for 2 independent experiments (CSS7 n = 5, CSS10 n = 6) (C57BL/6J n = 4, *Nfkbid*+/- n = 6); n.s., Mantel-Cox.
(TIF)

**S4 Fig. *Bumble* mice have intact T cell responses and normal cyst numbers during *Toxoplasma gondii* infections.** A) Peritoneal CD4 and CD8 T cells from *bumble* and C57BL/6J mice were assessed between days 32 and 35 of chronic infection with the type III CEP strain by an in vitro recall assay and assayed for intracellular IFNγ and IL-2. In brief, peritoneal cells were harvested and infected with live type I RH parasites for 16 hrs. T cells were assessed for production of IFNγ and IL-2 by intracellular staining and FACS. B) Brain cysts were enumerated between days 32–35 of chronic infection. C) As in A, but T cell recall responses were assessed on day 7 of secondary infection with the type I RH strain. D) Peritoneal T-regulatory cells (CD4+ CD25+ Foxp3+) were quantified on day 7 of secondary infection with type I RH strain. Each dot represents the result from one mouse, and plotted are cumulative averages

+SD from 3 experiments for C and D and 2 experiments for A and B; no significant differences were observed between *bumble* and C57BL/6J mice by unpaired parametric t-tests for data in B-D and the IFNγ response in A; the IL-2 response in A was not significant by a non-parametric Mann Whitney t-test.
(TIF)

**S5 Fig. *Bumble* mice have decreased marginal zone and atypical B cells before and during *Toxoplasma gondii* infection.** A) Frequency of marginal zone (MZ) B cells (CD21+ CD23$^{mid}$) among total splenic CD19+ B220+ B cells, and B) frequency of atypical B cells (FCRL5+ CD80 +) among total splenic CD19+ B220+ CD23+ CD21$^{mid}$ CD73+ memory B cells, at naïve, d12 of primary infection, and chronic infection with the type III strain in *bumble* and C57BL/6J mice. Cumulative data from two experiments n = 5–6 mice/condition. Significance was assessed with an unpaired two-tailed t-test; **** P<0.0001.
(TIF)

**S6 Fig. Assessing PerC reconstitutions by serum IgM ELISAs and flow cytometry, and survival of muMT mice.** A) Serum IgM from C57BL/6J, muMT, *bumble*, and A/J mice was measured by ELISA. Serum was harvested from mice either naïve, chronically infected with the type III CEP strain, or on D5 post-secondary infection with the type I GT1 *T. gondii* strain. PerC transfer (+) refers to mice adoptively transferred 5x10$^6$ total C57BL/6J PerC cells as a day 2 neonate. Each dot represents the results from an individual mouse, and plotted is the average +/-SD of the O.D. obtained at 450nm; *P<0.05, unpaired two-tailed t-test. B) *Bumble* reconstitution of the peritoneal B-1 compartment after neonatal PerC adoptive transfer. Representative FACS plots of peritoneal B-2 cells (B220$^{high}$ CD19+) and B-1 (B220$^{int-neg}$ CD19+) cells from *bumble* mice with or without PerC adoptive transfer. Shown are mice on day 20 of primary infection with the type III CEP strain. C) B cell deficient muMT mice (n = 3), muMT given WT PerC adoptive transfers as 2-day neonates then allowed to reconstitute for 6–7 weeks into adulthood (n = 2), and muMT given B cell enriched splenocytes (n = 3) 1 day prior to infection with the type III CEP strain were assessed for survival. D) muMT reconstitution of the B-2 cell compartment. WT and muMT with B cell enriched splenocytes (EasySep Mouse Pan-B Cell Isolation Kit, cat# 19844) adoptively transferred 24 hrs earlier, representative FACS plots of peritoneal B-2 cells (B220$^{high}$ CD19+) and B-1 (B220$^{int-neg}$ CD19+) are shown. E) muMT reconstitution of peritoneal B cell compartment after neonatal adoptive transfer. Representative FACS plots of peritoneal B-2 cells (B220$^{high}$ CD19+) and B-1 B cells (B220$^{int-neg}$ CD19+) from WT, and muMT mice or muMT mice with neonatal PerC adoptive transfer. For D and E, uninfected mice are 6–8 weeks of age and numbers indicate the percent of cells that fall within the depicted gate.
(TIF)

**S7 Fig. Bone marrow chimeric mice fail to survive primary infection but exhibit improved survival to vaccine strains.** A) Survival of the indicated bone marrow (BM) chimeras infected with the type III CEP *T. gondii* strain are plotted from a single experiment (n = 2 for *bumble* recipients per condition; n = 1 for C57BL/6J recipients); n.s., not significant, Mantel-Cox. B) Survival of the indicated BM chimeras vaccinated (10$^6$ i.p.) with the uracil auxotroph strain, Rh *Δup Δompdc*. Results are cumulative from 2–3 separate transfers and vaccinations; (n = 4–9 per condition); * P<0.05, Mantel-Cox.
(TIF)

**S8 Fig. Enhanced frequencies of peritoneal IgM- IgD- B-1 cells in resistant A/J mice following *Toxoplasma gondii* infection.** Frequencies of peritoneal (PerC) B-1a (CD5+) or B-1b (CD5-) B-1 cells (CD19+ B220$^{int-neg}$ CD43+) that are IgM+IgD- or IgM-IgD- in A/J and

C57BL/6J mice at the indicated infection states. The cumulative average +SD from 2–4 experiments are plotted and each dot represents the result from an individual mouse; P values calculated by 2-way ANOVA with Tukey correction; **** $P < 0.0001$, ** $P < 0.01$, * $P < 0.05$.
(TIF)

**S9 Fig. CD8 T cell IFNγ frequencies are increased in resistant A/J mice.** A) Peritoneal, and B) splenic cells were harvested from A/J and C57BL6/J mice chronically infected with the type III CEP *T. gondii* strain, and infected with live type I parasites for 16 hrs. T cells were assessed for production of granzyme B (GZB), IFNγ, and IL-2 by intracellular staining and FACS. The average frequency +/-SD of positive staining CD4+ or CD8+ T cells (CD3+ CD19-) and cumulative results from 2–3 experiments (AJ n = 8, C57BL6/J n = 11) are shown; * $P < 0.05$, unpaired two-tailed t-test.
(TIF)

**S10 Fig. Genes uniquely induced in B-1b cells on day 5 of secondary infection in resistant A/J mice.** Transcriptomic analysis of peritoneal B-1a (CD19+ B220^int-neg CD11b+ CD5+), B-1b (CD19+ B220^int-neg CD11b+ CD5-) and B-2 (CD19+ B220^hi CD11b- CD5-) B cells from A/J and C57BL/6J mice was performed using 3'-Tag RNA sequencing. A) Genes that were differentially upregulated in B-1b cells on day 5 of secondary infection compared to naïve mice in the A/J genetic background. P values of differentially expressed genes were calculated using the Benjamini-Hochberg adjustment for false discovery rate, and only those genes that survived significance were included in the heatmap. For comparison, all B-1 compartments in A/J mice are shown for this gene set. A/J B-1 Pathway and GO term enrichment was assessed on the genes presented in the heatmap in A. P values for enrichment analysis were adjusted with the Holm-Bonferroni correction. B) A cluster of genes found to be differentially induced in B-1b cells in A/J compared to C57BL/6J mice on day 5 of secondary infection are plotted as a heat map. C) Gene set enrichment analysis of the rank-ordered list of differentially expressed genes between A/J and C57BL/6J B-1b cells at D5 of secondary infection. Gene set depicted was in the top 10 gene sets ranked by false discovery rate (FDR) after investigating MSigDB's C7: immunologic signatures collection. Enrichment score is the degree of overrepresentation of a gene set at the top or bottom of a ranked list. NES is the enrichment score after normalizing for gene set size.
(TIF)

**S11 Fig. *Nfkbid* gene expression controls humoral responses to *Toxoplasma gondii*.** C57BL/6 (B6) mice have high expression of *Nfkbid* and exhibit B cell responses to *T. gondii* that produce parasite-specific IgG1 and IgM. In contrast, A/J and 'Bumble het' mice (C57BL/6J x *bumble* F1; *Nfkbid*+/-) have lower *Nfkbid* expression and exhibit enhanced IgG1 responses to *T. gondii*. In the case of 'Bumble hets', the enhanced IgG1 response correlates with increased plasma blast differentiation. In the case of A/J mice, overall increased B-1 and B-2 cell responses are observed, though the exact role that *Nfkbid* plays in B cells in this genetic background is currently undetermined. Finally, 'Bumble' mice do not express *Nfkbid* and have overall poor humoral immunity to *T. gondii*. Bumble mice produce no parasite-specific IgM and have greatly reduced parasite-specific IgG. There is also a defect in B cell maturation during chronic infection in Bumble mice. We hypothesize that intermediate expression of *Nfkbid* represents the optimal level for humoral immunity to *T. gondii*. Schematic created with BioRender.com.
(TIF)

**S1 Table. Virulent strains of *Toxoplasma gondii* superinfect genetically resistant hosts.** CEP *hxgprt-* chronically infected C57BL/10, C57BL/10.A, and A/J mice were given a secondary

infection with the indicated atypical *T. gondii* strain; then, 35–45 days after secondary infection, brains from surviving mice were homogenized in PBS and used to inoculate HFF monolayers. The resultant parasite cultures were then tested for the presence of the secondary infection strain; numerator = number of mice tested positive for secondary infection strain in the brain; denominator = number of mice that exhibited parasite growth in the HFF monolayer prior to drug selection. [1]Parasite growth was observed in mycophenolic acid / xanthine medium, which selects for parasites encoding a functional *HXGPRT* gene (i.e. the challenging atypical strains) and kills the primary infection CEP *hxgprt-* strain.
(XLSX)

**S2 Table. Primary and secondary infection survival of recombinant inbred mice (AxB; BxA).** 26 strains (n = 2) from the RI (AxB; BxA) panel were infected with $10^4$ type III avirulent CEP *hxgprt- T. gondii* parasites; then, 35 days later, mice were challenged with $5 \times 10^4$ virulent type I GT1 *T. gondii* parasites. Primary and secondary infection survival percentages are indicated. 35–45 days after secondary challenge, brains from the surviving RI mice were homogenized in PBS and used to inoculate HFF monolayers. The resultant parasite cultures were then tested for the presence of the challenging strain; numerator = number of mice tested positive for secondary infection strain in the brain; denominator = number of mice that exhibited parasite growth in the HFF monolayer prior to drug selection. Some surviving mice failed to generate parasite positive cultures. [1]Parasite growth was observed in mycophenolic acid / xanthine medium, which selects for GT1 parasites (the challenging strain) encoding the endogenous *HXGPRT* gene and against the primary infection CEP *hxgprt-* strain.
(XLSX)

**S1 Dataset. Polymorphic genes between A/J and C57BL/6J encoded within the four QTLs that define immunity to *Toxoplasma gondii*.** For each of the four QTLs that define secondary infection immunity to *T. gondii*, a list of all genes that have a DNA polymorphism between A/J and C5BL/6J mice, and are encoded within the QTL boundaries defined by the maximal genetic marker (real or imputed markers inferred in r/QTL) and the two flanking markers on either side; LOD scores and position are indicated. For each gene, its location and unique identifiers are listed, as well as the number and class of small nucleotide polymorphism (SNP) between A/J and C57BL/6J are shown. Total number of SNPs for each gene were also tallied ('Sum mutation'), and in bold are genes whose SNPs are two standard deviations greater than the average SNP for that class within the QTL region. In these QTL regions, there were also 476 (chr7), 88 (chr10), 40 (chr11) and 375 SNPs (chr17) that did not associate with any gene in the QTL boundaries. UTR, untranslated region; 'Region SNP', SNPs that are +/- 2 kb of the gene boundaries. Data obtained from Mouse Genome Informatics (MGI) (informatics.jax.org), and each locus is in a separate tab.
(XLSX)

**S2 Dataset. Transcriptomic analysis of B cells responding to *Toxoplasma gondii* infection.** Table of genes included (threshold of > = 5 CPM in any sample) in differential gene expression analysis. For each gene the Ensembl ID, MGI ID, gene symbol and descriptions are listed. The CPM from 3 samples of peritoneal B-1a, B-1b, and B-2 at naïve (N), chronic (Ch), and day 5 (D5) of secondary infection in both A/J (AJ) and C57BL/6J (B6) mice are given. The average CPM was calculated for each set of samples and average counts less than 5 CPM were raised to 5 to reduce the effect of low expression genes in further analysis. The log2 fold change (FC) of gene expression was calculated using the average CPM for each cell type (B-1a, B-1b, and B-2) of each strain (A/J and C57BL/6J) as they progressed from naïve to chronic infection or chronic to D5 secondary infection. The Log2 FC was also calculated from naïve to D5 of

secondary infection. The Log2 FC between A/J and C57BL/6J was calculated for each cell type and time point. Genes were then ranked by the number of comparisons that exhibited greater than 4FC by the indicated comparison.
(XLSX)

## Acknowledgments

We would like to thank David Gravano and the UC Merced Stem Cell Instrumentation Foundry for their assistance designing panels and help with flow cytometry. We thank John Boothroyd (Stanford University) for the rabbit anti-GRA7 antibody.

## Author Contributions

**Conceptualization:** Scott P. Souza, Nicole Baumgarth, Kirk D. C. Jensen.

**Data curation:** Scott P. Souza, Samantha D. Splitt, Juan C. Sànchez-Arcila.

**Formal analysis:** Scott P. Souza, Samantha D. Splitt, Juan C. Sànchez-Arcila, Julia A. Alvarez, Kirk D. C. Jensen.

**Funding acquisition:** Kirk D. C. Jensen.

**Investigation:** Scott P. Souza, Samantha D. Splitt, Juan C. Sànchez-Arcila, Julia A. Alvarez, Jessica N. Wilson, Safuwra Wizzard, Kirk D. C. Jensen.

**Methodology:** Scott P. Souza, Samantha D. Splitt, Juan C. Sànchez-Arcila, Julia A. Alvarez, Jessica N. Wilson, Safuwra Wizzard, Nicole Baumgarth, Kirk D. C. Jensen.

**Project administration:** Scott P. Souza, Kirk D. C. Jensen.

**Resources:** Zheng Luo, Nicole Baumgarth, Kirk D. C. Jensen.

**Supervision:** Kirk D. C. Jensen.

**Validation:** Scott P. Souza, Samantha D. Splitt, Juan C. Sànchez-Arcila, Julia A. Alvarez, Kirk D. C. Jensen.

**Visualization:** Scott P. Souza, Samantha D. Splitt, Juan C. Sànchez-Arcila, Julia A. Alvarez, Jessica N. Wilson, Kirk D. C. Jensen.

**Writing – original draft:** Scott P. Souza, Kirk D. C. Jensen.

**Writing – review & editing:** Scott P. Souza, Nicole Baumgarth, Kirk D. C. Jensen.

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
