## [Decision Letter · Decision Letter 0]

7 Jul 2021

Dear Prof Jensen,

Thank you very much for submitting your manuscript "Nfkbid is required for immunity and antibody responses to Toxoplasma gondii" for consideration at PLOS Pathogens. As with all papers reviewed by the journal, your manuscript was reviewed by members of the editorial board and by several independent reviewers. In light of the reviews (below this email), we would like to invite the resubmission of a significantly-revised version that takes into account the reviewers' comments.

Reviewer 1 in particular raises several points that the authors will need to address in a revision manuscript.

We cannot make any decision about publication until we have seen the revised manuscript and your response to the reviewers' comments. Your revised manuscript is also likely to be sent to reviewers for further evaluation.

Sincerely,

Eric Y Denkers

Associate Editor

PLOS Pathogens

Kami Kim

Section Editor

PLOS Pathogens

Kasturi Haldar

Editor-in-Chief

PLOS Pathogens

orcid.org/0000-0001-5065-158X

Michael Malim

Editor-in-Chief

PLOS Pathogens

orcid.org/0000-0002-7699-2064

Reviewer 1 in particular raises several points that the authors will need to address in a revision manuscript.

Reviewer's Responses to Questions

**Part I - Summary**

Reviewer #1: In this manuscript, Souza et al analyze the mechanisms by which mice previously exposed to T. gondii resist a secondary challenge with a more virulent, heterologous, strain. They take advantage of a genetic dissection in mice using susceptible (C57BL/6) vs. resistant (A/J) mice, and they study more specifically the implication in this phenotype of Nfkbid and of different B cell subsets.

This paper addresses an important but largely neglected aspect of anti-T. gondii immunity, namely the contribution of antibody responses in protective immunity upon secondary parasite exposure. It also sheds light on a relatively poorly studied atypical inhibitor of NF-kB (Nfkbid gene encoding for IkBNS), in the context T. gondii infection. Using multiple models of mice and parasite strains, the authors report intriguing and original findings, and the study is clearly the reflection of a tremendous amount of work. However, due to the fact that certain set-ups fluctuate throughout the figures and that some data are somewhat paradoxical (see below), I must say I had a hard time tying all pieces together in a way that made complete sense. The use of a full Nfkbid KO model also comes with obvious limitations.

At this stage, my feeling is that the manuscript has potential but some reformulating/reshuffling, a few additional controls (see below), and a schematics of the working model, would be valuable to clarify and enforce the message.

Reviewer #2: This study by Souza, Jensen, Baumgarth and their colleagues highlights the complexity of immunological elements that are required for strain-heterologous immunity to parasites, when the priming and challenge strains vary in their levels of virulence and potentially antigen display. Through their thorough analysis of the genetic architecture that controls immune resistance to secondary challenge with virulent strains of T. gondii, they have exposed the underlying layers of complex additive interactions between multiple gene loci and have highlighted the role of Nfkbid, a regulator of Nf-kB in regulating humoral immunity to T. gondii. Their rigorous analyses have revealed a cooperative role of B-1 and B2 lineage cells in providing protective humoral immune responses and have dissected their individual and combined roles at steady state and during the dynamics of the infection. Although their analysis of Nfkbid failed to account for the initial observed differences in genetically resistant and susceptible strains of mice, the study still is very significant, as it reveals novel aspects of B cell activation, synergy and dynamics that can only be observed with live parasite infection and challenge scenarios. This study will help spur further detailed and layered analyses of the B cell responses to T. gondii and perhaps other parasitic agents.

Reviewer #3: The strength of the manuscript is that it describes the role of Nfkbid in the development of B cell responses during T.gondii infection which has not been previously reported. The weakness is that the difference between C57BL/6 and A/j mice are not conclusive unless this molecule is over expressed or deleted and the response determined.

**Part II – Major Issues: Key Experiments Required for Acceptance**

Reviewer #1: 1. My 1st interrogation is about the contribution of neutralizing antibodies and the respective roles of B-1 vs B-2 cell subsets in resistance to secondary infection. Fig 4A and 4E suggest a beneficial function of B-1 but the RH tachyzoite coating ability of B-1-derived antibodies appears to be much lower than that of B-2-derived antibodies (Fig 4D). How do the authors reconcile these observations? They set-up a nice assay to detect binding by Facs but a caveat is that this assay does not quantify the neutralization capacity of the immunoglobulins. Could they evaluate the neutralization potential of C57BL/6- vs A/J-derived abs, or of B-1 vs B-2-derived abs, in a parasite invasion assay (as in Fig 3B)?

2. The authors argue that a B cell-intrinsic role of Nfkbid regulates resistance to secondary infection. However, in the absence of selective B cell depletion or B cell-specific conditional ablation of Nfkbid, the formal role of Nfkbid within B cells is hard to formally establish. One way could be to transfer peritoneal cells from bumble mice in the experiment of Fig 4A? If bumble PerC cells did not improve survival (contrary to WT), it would provide a more direct proof for the B cell-intrinsic role of Nfkbid. Mixed BM chimeras with bumble BM + B cell KO (e.g. JH-/-) BM could be useful as well to create a system where all B cells are Nfkbid-deficient while other cells are in majority WT, but I admit that these experiments are clearly a long shot.

3. A last important point would be to determine if the challenged mice are ‘equally’ chronically infected at the time of GT1/RH challenge. Survival is similar between WT and bumble mice but is there a difference in CEP parasite burden in the CNS? One could also imagine that clearance may occur in some mice infected with such a low virulence strain. In any case, knowing the ‘starting’ parasite load before challenge seems important for the following interpretations.

Reviewer #2: (No Response)

Reviewer #3: The key experiment is to overexposes the molecule in C57BL/6 mice and to delete it in A/J mice and then look at the phenotype.

**Part III – Minor Issues: Editorial and Data Presentation Modifications**

Reviewer #1: • Some experimental set-ups vary across the figures (e.g. RH vs. GT1 challenge, primary infection vs. vaccination). I can understand there are justifications for this but, at the same time, it makes it difficult to be sure that they are looking at the same phenomenon all throughout the manuscript. The authors have put a lot of efforts trying to define the determinants of resistance to a lethal secondary infection. But they encounter difficulties working with highly virulent strains in laboratory mice that are known to be very susceptible, which forced them to revise some of the set-ups (vaccinate instead of infect irradiated mice). In a sense, one is left to wonder what the contribution of humoral responses, B cell subsets, and Nfkbid, is in a more ‘simple’ setting, for example a challenge with a high dose of the same (CEP) strain. Such a more ‘simple’ approach would have also helped set the stage to better understand the more complex experiments described here. Perhaps these data already exist? If so, could they remind them to the reader?

• Unless I misunderstood, I see a disconnect between the genetic dissection pointing to Nfkbid, among several other genes, as a polymorphic gene potentially involved in A/J resistance, and the findings obtained with Nfkbid KO mice in a C57BL/6 background. On the one hand, Nfkbid is required for resistance in C57BL/6 but on the other hand, the naturally resistant A/J display higher levels of Nfkbid. Is it not a counterintuitive observation? I think such puzzling data would deserve to be discussed more thoroughly. If they have a reconciling hypothesis, it would also greatly help to illustrate it on a working model schematics.

• In my opinion, the title oversimplifies the data presented here. It does not convey the seemingly complex role of Nfkbid (more a regulator than a positively required element) and the fact that most experiments addressed the resistance to the acute phase of a secondary infection, rather than the general immune responses to T. gondii, or the control of parasite latency.

• Beside generating a humoral response, CEP primary infection likely induces memory T lymphocytes, which could also be involved in the resistance that is observed against secondary type I challenge. The intracellular cytokine stainings presented in Fig S3 are informative but in my opinion, they may be biased by strong differences in acute phase reported in the two models (Fig 2E), which likely lead to distinct levels of inflammatory cytokines and lymphocyte activation. To more formally exclude a role for T cells in the phenotype, I think it would be important to verify that the T cell responses elicited in the two groups are similar before challenge. Perhaps this could be done by restimulating peritoneal or spleen cells with CEP-infected DC, and/or with endogenous T cell antigens from T. gondii.

• I may have missed it but the justification for focusing on Nfkib after the genetic dissection is not obvious. What do the non-synonymous SNP exactly change to the protein? Do they introduce premature stop codons? What do they do with other candidate genes? Is their implication ruled out, how?

• Fig S1: inconsistency of symbol between legend and graph for B6.

• Lines 173, 308 & 335: misformatted ref

Statistics

• Fig S1. What does the regression value represent? Why not applying a statistical test to compare the samples?

• Fig 1D. In my understanding, applying a Fisher’s test on percentages is not appropriate. Rather than showing the fraction of mice superinfected or not, the authors could represent the actual number of mice in each category and then apply the Fisher’s exact test.

• Fig 2F: was the normality hypothesis verified to be able to use a parametric t-test?

• Fig S3. I do not understand why the authors made a correction for multiple comparisons. Is the purpose not to compare WT and bumble per cytokine? Same as above: was the normality hypothesis verified to use a parametric t-test?

Reviewer #2: (No Response)

Reviewer #3: None

PLOS authors have the option to publish the peer review history of their article (what does this mean?). If published, this will include your full peer review and any attached files.

Reviewer #1: No

Reviewer #2: No

Reviewer #3: **Yes: **Imtiaz A Khan
---

## [Decision Letter · Decision Letter 1]

1 Nov 2021

Dear Prof Jensen,

We are pleased to inform you that your manuscript 'Genetic mapping reveals Nfkbid as a central regulator of humoral immunity to Toxoplasma gondii' has been provisionally accepted for publication in PLOS Pathogens.

Best regards,

Eric Y Denkers

Associate Editor

PLOS Pathogens

Kami Kim

Section Editor

PLOS Pathogens

Kasturi Haldar

Editor-in-Chief

PLOS Pathogens

orcid.org/0000-0001-5065-158X

Michael Malim

Editor-in-Chief

PLOS Pathogens

orcid.org/0000-0002-7699-2064

Reviewer Comments (if any, and for reference):

Reviewer's Responses to Questions

**Part I - Summary**

Reviewer #1: The authors addressed all my comments in a thorough and convincing way. Thank you, I have no other concern regarding this manuscript.

Reviewer #3: (No Response)

**Part II – Major Issues: Key Experiments Required for Acceptance**

Reviewer #1: (No Response)

Reviewer #3: (No Response)

**Part III – Minor Issues: Editorial and Data Presentation Modifications**

Reviewer #1: (No Response)

Reviewer #3: (No Response)

PLOS authors have the option to publish the peer review history of their article (what does this mean?). If published, this will include your full peer review and any attached files.

Reviewer #1: No

Reviewer #3: No

---

## [Editor Report · Acceptance letter]

30 Nov 2021

Dear Prof Jensen,

We are delighted to inform you that your manuscript, "Genetic mapping reveals Nfkbid as a central regulator of humoral immunity to Toxoplasma gondii," has been formally accepted for publication in PLOS Pathogens.

Best regards,

Kasturi Haldar

Editor-in-Chief

PLOS Pathogens

orcid.org/0000-0001-5065-158X

Michael Malim

Editor-in-Chief

PLOS Pathogens

orcid.org/0000-0002-7699-2064